# Matrix Inference and Estimation in Multi-Layer Models

**Parthe Pandit**[*]
Dept. ECE
UC, Los Angeles
parthepandit@ucla.edu

**Mojtaba Sahraee-Ardakan**
Dept. ECE
UC, Los Angeles
msahraee@ucla.edu

**Sundeep Rangan**
Dept. ECE
NYU
srangan@nyu.edu

**Philip Schniter**
Dept. ECE
The Ohio State Univ.
schniter.1@osu.edu

**Alyson K. Fletcher**
Dept. Statistics
UC, Los Angeles
akfletcher@ucla.edu

## Abstract

We consider the problem of estimating the input and hidden variables of a stochastic multi-layer neural network from an observation of the output. The hidden variables in each layer are represented as matrices with statistical interactions along both rows as well as columns. This problem applies to matrix imputation, signal recovery via deep generative prior models, multi-task and mixed regression, and learning certain classes of two-layer neural networks. We extend a recently-developed algorithm – Multi-Layer Vector Approximate Message Passing (ML-VAMP), for this matrix-valued inference problem. It is shown that the performance of the proposed Multi-Layer Matrix VAMP (ML-Mat-VAMP) algorithm can be exactly predicted in a certain random large-system limit, where the dimensions $N \times d$ of the unknown quantities grow as $N \to \infty$ with $d$ fixed. In the two-layer neural-network learning problem, this scaling corresponds to the case where the number of input features as well as training samples grow to infinity but the number of hidden nodes stays fixed. The analysis enables a precise prediction of the parameter and test error of the learning.

## 1   Introduction

Consider an $L$-layer stochastic neural network given by

$$\mathbf{Z}_\ell^0 = \mathbf{W}_\ell \mathbf{Z}_{\ell-1}^0 + \mathbf{B}_\ell + \mathbf{\Xi}_\ell^0, \qquad \ell = 1, 3, \ldots, L-1, \tag{1a}$$

$$\mathbf{Z}_\ell^0 = \boldsymbol{\phi}_\ell(\mathbf{Z}_{\ell-1}^0, \mathbf{\Xi}_\ell^0), \qquad \ell = 2, 4, \ldots, L, \tag{1b}$$

where, for $\ell = 0, 1, \ldots, L$, we have *true* activations $\mathbf{Z}_\ell^0 \in \mathbb{R}^{n_\ell \times d}$, weights $\mathbf{W}_\ell \in \mathbb{R}^{n_\ell \times n_{\ell-1}}$, bias matrices $\mathbf{B}_\ell \in \mathbb{R}^{n_\ell \times d}$, and *true* noise realizations $\mathbf{\Xi}_\ell^0$. The activation functions $\boldsymbol{\phi}_\ell : \mathbb{R}^{n_{\ell-1} \times d} \to \mathbb{R}^{n_\ell \times d}$ are known non-linear functions acting row-wise on their inputs. See Fig. 1 (TOP). We use the superscript $^0$ in $\mathbf{Z}_\ell^0$ to indicate the true values of the variables, in contrast to estimated values denoted by $\widehat{\mathbf{Z}}_\ell$ discussed later. We model the true values $\mathbf{Z}_0^0$ as a realization of random $\mathbf{Z}_0$, where the rows $\mathbf{z}_{0,i:}^\mathsf{T}$ of $\mathbf{Z}_0$ are i.i.d. with distribution $p_0$: $p(\mathbf{Z}_0) = \prod_{i=1}^{n_0} p_0(\mathbf{z}_{0,i:})$. Similarly, we also assume that $\mathbf{\Xi}_\ell^0$ are realizations of random $\mathbf{\Xi}_\ell$ with i.i.d. rows $\boldsymbol{\xi}_{\ell,i:}^\mathsf{T}$. For odd $\ell$, the rows $\boldsymbol{\xi}_{\ell,i:}$ are zero-mean multivariate Gaussian with covariance matrix $\mathbf{N}_\ell^{-1} \in \mathbb{R}^{d \times d}$, whereas for even $\ell$, the rows $\boldsymbol{\xi}_{\ell,i:}$ can be arbitrarily distributed but i.i.d.

---

[*]Code available at https://github.com/parthe/ML-Mat-VAMP

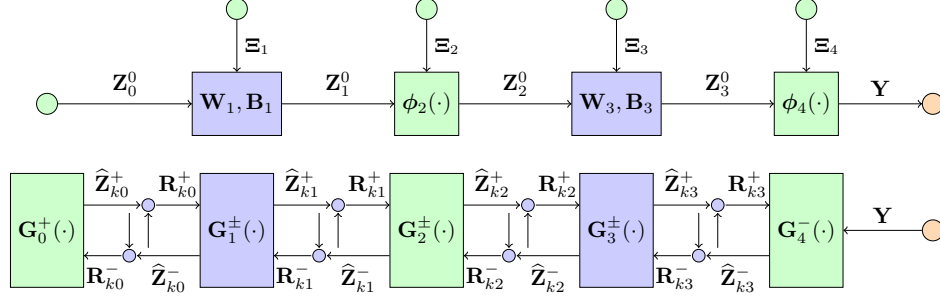

Figure 1: (TOP) The signal flow graph for *true* values of matrix variables $\{\mathbf{Z}_\ell^0\}_{\ell=0}^3$, given in eqn. (1) where $\mathbf{Z}_\ell^0 \in \mathbb{R}^{n_\ell \times d}$. (BOTTOM) Signal flow graph of the ML-MVAMP procedure in Algo. 1. The variables with superscript + and - are updated in the forward and backward pass respectively. ML-MVAMP (Algorithm 1) solves (2) by solving a sequence of simpler estimation problems over consecutive pairs $(\mathbf{Z}_\ell, \mathbf{Z}_{\ell-1})$.

Denoting by $\mathbf{Y} := \mathbf{Z}_L^0 \in \mathbb{R}^{n_L \times d}$ the output of the network, we consider the following matrix inference problem:

$$\text{Estimate } \mathbf{Z} := \{\mathbf{Z}_\ell\}_{\ell=0}^{L-1} \qquad \text{given } \mathbf{Y} := \mathbf{Z}_L^0 \text{ and } \{\mathbf{W}_{2k-1}, \mathbf{B}_{2k-1}, \phi_{2k}\}_{k=1}^{L/2}. \tag{2}$$

A key feature of the problem we consider here is that the unknowns, $\mathbf{Z}_\ell$, are *matrix-valued* with $d$ columns with statistical dependencies between the columns. As we will see in Section 2, the matrix-valued case applies to several problems of broad interest such as matrix imputation, multi-task and mixed regression problems, sketched clustering. We also show that via this formulation we can analyze the learning in two layer neural networks under some architectural assumptions.

In many applications, the inference problem can be performed via minimization of an appropriate cost function. For example, suppose the network (1) has no noise $\mathbf{\Xi}_\ell$ for all layers except the final measurement layer, $\ell = L$. In this case, the $\mathbf{Z}_{L-1}^0 = \mathbf{g}(\mathbf{Z}_0^0)$ for some *deterministic function* $\mathbf{g}(\cdot)$ representing the action of the first $L-1$ layers. Inference can then be conducted via a minimization of the form,

$$\widehat{\mathbf{Z}}_{L-1} := \mathbf{g}\left(\arg\min_{\mathbf{Z}_0} H_L(\mathbf{Y}, \mathbf{Z}_{L-1}) + H_0(\mathbf{Z}_0), \quad \text{subject to } \mathbf{Z}_{L-1} = \mathbf{g}(\mathbf{Z}_0)\right) \tag{3}$$

where the term $H_L(\mathbf{Y}, \mathbf{Z}_{L-1})$ penalizes the prediction error and $H_0(\mathbf{Z}_0)$ is an (optional) regularizer on the network input. For maximum a posteriori (MAP) estimation one takes, $H_L(\mathbf{Y}, \mathbf{Z}_{L-1}) = -\log p(\mathbf{Y}|\mathbf{Z}_{L-1})$, and $H_0(\mathbf{Z}_0) = -\log p(\mathbf{Z}_0)$, where the output probability $p(\mathbf{Y}|\mathbf{Z}_{L-1})$ is defined from the last layer of model (1b): $\mathbf{Y} = \mathbf{Z}_L = \phi_L(\mathbf{Z}_{L-1}, \mathbf{\Xi}_L)$. The minimization (3) can then be solved using a gradient-based method. Encouraging results in image reconstruction have been demonstrated in [46, 4, 15, 18, 38, 42, 29]. Markov-chain Monte Carlo (MCMC) algorithms and Langevin diffusion [7, 45] could also be employed for more complex inference tasks.

However, rigorous analysis of these methods is difficult due to the non-convex nature of the optimization problem. To address this issue, recent works [25, 12, 35] have extended Approximate Message Passing (AMP) methods to provide inference algorithms for the multi-layer networks. AMP was originally developed in [9, 10, 3, 17] for compressed sensing. Similar to other AMP-type results, the performance of multi-layer AMP-based inference can be precisely characterized in certain high-dimensional random instances. In addition, the mean-squared error for inference of the algorithms match predictions for the Bayes-optimal inference predicted by various techniques from statistical physics [37, 14, 2]. Thus, AMP-based multi-layer inference provides a computationally tractable estimation framework with precise performance guarantees and testable conditions for optimality in certain high-dimensional random settings.

Prior multi-layer AMP works [16, 26, 12, 35] have considered the case of vector-valued quantities with $d = 1$. The main contribution of this paper is to consider the *matrix-valued* case when $d > 1$. To handle the case when $d > 1$, we extend the Multi-Layer Vector Approximate Message Passing (ML-VAMP) algorithm of [12, 35] to the matrix case. The ML-VAMP method is based on VAMP method of [36], which is closely related to expectation propagation (EP) [28, 39], expectation-

consistent approximate inference (EC) [32, 13], S-AMP [6], and orthogonal AMP [24]. We will use "ML-Mat-VAMP" when referring to the matrix extension of ML-VAMP.

**Contributions:** First, similar to the case of ML-VAMP, we analyze ML-Mat-VAMP in a large system limit, where $n_\ell \to \infty$ and $d$ is fixed, under rotationally invariant random weight matrices $\mathbf{W}_\ell$. In this large system limit, we prove that the mean-squared error (MSE) of the estimates of ML-Mat-VAMP can be exactly predicted by a deterministic set of equations called the *state evolution* (SE). The SE describes how the distribution of the true activations and pre-activations of the network as well as the estimated values generated by ML-Mat-VAMP evolve jointly from one iteration of the algorithm to the other. This extension of the SE equations to the matrix case is not trivial and requires considering correlation across multiple vectors. Indeed, in the case of ML-VAMP, the SE equations involve scalar quantities and $2 \times 2$ matrices. For ML-Mat-VAMP, the SE equations involve $d \times d$ and $2d \times 2d$ matrices.

Second, we show that the method can offer precise predictions in important estimation problems that are difficult to analyze via other means. The ML-VAMP was focused on deep reconstruction problems [46, 4]. The matrix version here can be applied to other classes of problems such as multi-task regression, matrix completion and learning the input layer of a neural network. Even though these networks are typically shallow (just $L = 2$ layers), there are no existing methods that can provide the same types of precise results. For example, in the case of learning the input layer of a neural network, our results can exactly predict the test error as a function of the noise statistics, activations, number of training sample and other key modeling parameters.

**Notation:** Boldface uppercase letters $\mathbf{X}$ denote matrices. $\mathbf{X}_{n:}$ refers to the $n^{\text{th}}$ row of $\mathbf{X}$. Random vectors are row-vectors. For a function $f : \mathbb{R}^{1 \times m} \to \mathbb{R}^{1 \times k}$, its row-wise extension is represented by $\mathbf{f} : \mathbb{R}^{N \times m} \to \mathbb{R}^{N \times k}$, i.e., $[\mathbf{f}(\mathbf{X})]_{n:} = f(\mathbf{X}_{n:})$. We denote the Jacobian matrix of $f$ by $\frac{\partial f}{\partial \boldsymbol{x}}(\boldsymbol{x}) \in \mathbb{R}^{m \times k}$, so that $[\frac{\partial f}{\partial \boldsymbol{x}}(\boldsymbol{x})]_{ij} = \frac{\partial f_i}{\partial \boldsymbol{x}_j}(\boldsymbol{x})$. For its row-wise extension $\mathbf{f}$, we denote by $\langle \frac{\partial \mathbf{f}}{\partial \mathbf{X}}(\mathbf{X}) \rangle$ the average Jacobian, i.e., $\frac{1}{N} \sum_{n=1}^{N} \frac{\partial f}{\partial \mathbf{X}_{n:}}(\mathbf{X}_{n:}) \in \mathbb{R}^{m \times k}$

## 2 Example Applications

As we describe next, the matrix estimation problem 2 is of broad interest and several interesting applications can be formulated under this framework. We share a few examples below.

### 2.1 Multi-task and Mixed Regression Problems

A simple application of the matrix-valued multi-layer inference problem (2) is for *multi-task regression* [31]. Consider a generalized linear model of the form,

$$\mathbf{Y} = \phi(\mathbf{X}\mathbf{F}; \boldsymbol{\Xi}), \tag{4}$$

where $\mathbf{Y} \in \mathbb{R}^{N \times d}$ is a matrix of measured responses, $\mathbf{X} \in \mathbb{R}^{N \times p}$ is a known design matrix, $\mathbf{F} \in \mathbb{R}^{p \times d}$ are a set regression coefficients to be estimated, and $\boldsymbol{\Xi}$ is noise. The problem can be considered as $d$ separate regression problems – one for each column. However, in some applications, these design "tasks" are related in such a way that it benefits to *jointly* estimate the predictors. To do this, it is common to solve an optimization problem of the form

$$\arg\min_{\mathbf{F}} \left\{ \sum_{j=1}^{d} \sum_{i=1}^{N} L(y_{ij}, [\mathbf{X}\mathbf{F}]_{ij}) + \lambda \sum_{k=1}^{p} \rho(\mathbf{F}_{k:}) \right\}, \tag{5}$$

where $L(\cdot)$ is a loss function, and $\rho(\cdot)$ is a regularizer that acts on the rows $\mathbf{F}_{k:}$ of $\mathbf{F}$ to couple the prediction coefficients across tasks. For example, the multi-task LASSO [31] uses loss $L(y, z) = (y - z)^2$ and regularization $\rho(\mathbf{F}_{k:}) = \|\mathbf{F}_{k:}\|_2$ to enforce row-sparsity in $\mathbf{F}$. In the compressive-sensing context, multi-task regression is known as the "multiple measurement vector" (MMV) problem, with applications in MEG reconstruction [8], DoA estimation [43], and parallel MRI [22]. An AMP approach to the MMV problem was developed in [48]. The multi-task model (4) can be immediately written as a multi-layer network (1) by setting: $\mathbf{Z}_0 := \mathbf{F}, \mathbf{W}_0 := \mathbf{X}, \mathbf{Z}_1 := \mathbf{W}_0 \mathbf{Z}_0 = \mathbf{X}\mathbf{F}, \mathbf{Y} = \mathbf{Z}_2 := \phi(\mathbf{Z}_1, \boldsymbol{\Xi})$. Also, by appropriately setting the prior $p(\mathbf{Z}_0)$, the multi-layer matrix MAP inference (3) will match the multi-task optimization (5).

In (5), the regularization couples the columns of $\mathbf{F}$ but the loss term couples its rows. In *mixed regression* problems, the loss couples the columns of $\mathbf{F}$. For example, consider designing predictors $\mathbf{F} = [\mathbf{f}_1, \mathbf{f}_2]$ for *mixed linear regression* [47], i.e.,

$$y_i = q_i \mathbf{x}_i^\mathsf{T} \mathbf{f}_1 + (1 - q_i) \mathbf{x}_i^\mathsf{T} \mathbf{f}_2 + v_i, \quad q_i \in \{0, 1\}, \tag{6}$$

where $i = 1, \ldots, N$ and the $i$th response comes from one of two linear models, but which model is not known. This setting can be modeled by a different output mapping: As before, set $\mathbf{Z}_0 := \mathbf{F}$, $\mathbf{Z}_1 = \mathbf{X}\mathbf{F}$ and let the noise in the output layer be $\mathbf{\Xi}_1 = [\mathbf{q}, \mathbf{v}]$ which includes the additive noise $v_i$ in (6) and the random selection variable $q_i$. Then, we can write (6) via an appropriate function, $\mathbf{y} = \phi_1(\mathbf{Z}_1, \mathbf{\Xi}_1)$.

## 2.2 Sketched Clustering

A related problem arises in *sketched clustering* [19], where a massive dataset is nonlinearly compressed down to a short vector $\mathbf{y} \in \mathbb{R}^n$, from which cluster centroids $\mathbf{f}_k \in \mathbb{R}^p$, for $k = 1, \ldots, d$, are then extracted. This problem can be approached via the optimization [20] $\min_{\boldsymbol{\alpha} \geq \mathbf{0}} \min_{\mathbf{F}} \sum_{i=1}^n \left| y_i - \sum_{j=1}^d \alpha_j e^{\sqrt{-1} \mathbf{x}_i^\mathsf{T} \mathbf{f}_j} \right|^2$ where $\mathbf{x}_i \in \mathbb{R}^p$ are known i.i.d. Gaussian vectors. An AMP approach to sketched clustering was developed in [5]. For known $\boldsymbol{\alpha}$, the minimization corresponds to MAP estimation with the multi-layer matrix model with $\mathbf{Z}_0 = \mathbf{F}$, $\mathbf{W}_1 = \mathbf{X}$ $\mathbf{Z}_1 = \mathbf{X}\mathbf{F}$ and using the output mapping, $\phi_1(\mathbf{Z}_1, \mathbf{\Xi}) := \sum_{j=1}^d \alpha_j e^{\sqrt{-1} \mathbf{Z}_{1,:j}} + \mathbf{\Xi}$, where the exponential is applied elementwise and $\mathbf{\Xi}$ is i.i.d. Gaussian. The mapping $\phi_1$ operates row-wise on $\mathbf{Z}_1$ and $\mathbf{\Xi}$.

## 2.3 Learning the Input Layer of a Two-Layer Neural Network

The matrix inference problem (2) can also be applied to learning the input layer weights in a two-layer neural network (NN). Let $\mathbf{X} \in \mathbb{R}^{N \times N_{\text{in}}}$ and $\mathbf{Y} \in \mathbb{R}^{N \times N_{\text{out}}}$ be training data corresponding to $N$ data samples. Consider the two-layer NN model,

$$\mathbf{Y} = \sigma(\mathbf{X}\boldsymbol{F}_1)\boldsymbol{F}_2 + \mathbf{\Xi}, \tag{7}$$

with weight matrices $(\boldsymbol{F}_1, \boldsymbol{F}_2)$, componentwise activation function $\sigma(\cdot)$, and noise $\mathbf{\Xi}$. In (7), the bias terms are omitted for simplicity. We used the notation "$\boldsymbol{F}_\ell$" for the weights, instead of the standard notation "$\boldsymbol{W}_\ell$," to avoid confusion when (7) is mapped to the multi-layer inference network (2). Now, our critical assumption is that the weights in the second layer, $\boldsymbol{F}_2$, are known. The goal is to learn only the weights of the first layer, $\boldsymbol{F}_1 \in \mathbb{R}^{N_{\text{in}} \times N_{\text{hid}}}$, from a dataset of $N$ samples $(\mathbf{X}, \mathbf{Y})$.

If the activation is ReLU, i.e., $\sigma(\boldsymbol{H}) = \max\{\boldsymbol{H}, 0\}$ and $\mathbf{Y}$ has a single column (i.e. scalar output per sample), and $\boldsymbol{F}_2$ has all positive entries, we can, without loss of generality, treat the weights $\boldsymbol{F}_2$ as fixed, since they can always be absorbed into the weights $\boldsymbol{F}_1$. In this case, $\mathbf{y}$ and $\mathbf{F}_2$ are vectors and we can write the $i$th entry of $\mathbf{y}$ as

$$y_i = \sum_{j=1}^d F_{2j} \sigma([\mathbf{X}\boldsymbol{F}_1]_{ij}) + \xi_i = \sum_{j=1}^d \sigma([\mathbf{X}\boldsymbol{F}_1]_{ij} F_{2j}) + \xi_i \tag{8}$$

Thus, we can assume, without loss of generality, that $\mathbf{F}_2$ is all ones. The parameterization (8) is sometimes referred to as the *committee machine* [41]. The committee machine has been recently studied by AMP methods [1] and mean-field methods [27] as a way to understand the dynamics of learning.

To pose the two-layer learning problem as multi-layer inference, define $\boldsymbol{Z}_0 := \boldsymbol{F}_1$, $\boldsymbol{W}_1 := \mathbf{X}$, $\boldsymbol{Z}_1 := \mathbf{X}\boldsymbol{F}_1$ $\mathbf{\Xi}_2 := \mathbf{\Xi}$, then $\boldsymbol{Y} = \boldsymbol{Z}_2$, where $\boldsymbol{Z}_2$ is the output of a 2-layer inference network of the form in (1):

$$\mathbf{Y} = \boldsymbol{Z}_2 = \phi_2(\boldsymbol{Z}_1, \mathbf{\Xi}_2) := \sigma(\boldsymbol{Z}_1)\boldsymbol{F}_2 + \mathbf{\Xi}_2. \tag{9}$$

Note that $\boldsymbol{W}_1$ is known. Also, since we have assumed that $\boldsymbol{F}_2$ is known, the function $\phi_2$ is known. Finally, the function $\phi_2$ is row-wise separable on both inputs. Thus, the problem of learning the input weights $\boldsymbol{F}_1$ is equivalent to learning the input $\boldsymbol{Z}_0$ of the network (9).

## 2.4 Model-Based Matrix completion

Consider an observed matrix $\mathbf{Y} = \mathbf{Z}_L \in \mathbb{R}^{N_L \times d}$ with missing entries $\Omega^c \in [N_L] \times [d]$. The problem is to impute the missing entries of $\mathbf{Y}$. This is an important problem in several applications ranging from recommendation systems, genomics, bioinformatics and more broadly analysis of tabular data. There have been several approaches to solving this data imputation problem, right from 0 imputation and mean imputation to more sophisticated techniques based on generative models.

Consider a generative model based on a multi-layer perceptron as in (1) such that the output $\mathbf{Z}_{L-1}$ models the uncorrupted data matrix. Then the imputation problem can be posed as the solution of the MAP optimization problem:

$$\underset{\{\mathbf{Z}_\ell\}_{\ell=0}^{L}}{\text{minimize}} \|\mathbf{Y} - \mathbf{Z}_{L-1}\|_\Omega^2 - \log \mathbb{P}(\mathbf{Z}_{L-1}, \mathbf{Z}_{L-2}, \ldots, \mathbf{Z}_0) \qquad (10)$$

where $\|\mathbf{Y} - \mathbf{Z}_{L-1}\|_\Omega^2 = \sum_{(i,j) \in \Omega} ((\mathbf{Y})_{ij} - (\mathbf{Z}_{L-1})_{ij})^2$. One can also similarly construct Bayes estimators such as $\mathbb{E}[\mathbf{Z}_{L-1}|\mathbf{Z}_L]$.

Traditional approaches to matrix completion have looked at regularized convex minimization schemes just like (10) where $-\log \mathbb{P}(\mathbf{Z}_{L-1}) = \|\mathbf{Z}_{L-1}\|_*$, which is the nuclear norm, or some other structure inducing convex norms. While the term $-\log \mathbb{P}(\ldots)$ in (10) can be thought of as a more general regularization term, this formulation allows for more general application problems with heterogeneous variables.

For example, in imputation of tabular data, it is often the case that some columns correspond to continuous valued variables, whereas other variables are discrete valued modeling Yes/No answers or count data. In such scenarios the $-\log \mathbb{P}(\mathbf{Z}_{L-1}, \ldots)$ allows more flexibility towards modeling using GLMs and other exponential family distributions for every column separately. One simple instance of (10) would be a generative model $-\log \mathbb{P}(\mathbf{Z}_{L-1}, \ldots, \mathbf{Z}_0)$ which is trained on some fully observed data $\mathbf{Z}_{L-1}$ using unsupervised learning methods such as Variational Autoencoders (VAE) and Generative Adversarial Networks (GAN).

# 3 Multi-layer Matrix VAMP

## 3.1 MAP and MMSE inference

Observe that the equations (1) define a Markov chain over these signals and thus the posterior $p(\mathbf{Z}|\mathbf{Z}_L)$ factorizes as $p(\mathbf{Z}|\mathbf{Z}_L) \propto p(\mathbf{Z}_0) \prod_{\ell=1}^{L-1} p(\mathbf{Z}_\ell|\mathbf{Z}_{\ell-1}) \, p(\mathbf{Y}|\mathbf{Z}_{L-1})$. where recall the notation $\mathbf{Z}$ from (2). The transition probabilities $p(\mathbf{Z}_\ell|\mathbf{Z}_{\ell-1})$ above are implicitly defined in equation (1) and depend on the statistics of noise terms $\mathbf{\Xi}_\ell$. We consider both maximum *a posteriori* (MAP) and minimum mean squared error (MMSE) estimation for this posterior:

$$\widehat{\mathbf{Z}}_{\mathsf{map}} = \arg \max_{\mathbf{Z}} \ p(\mathbf{Z}|\mathbf{Z}_L) \qquad \widehat{\mathbf{Z}}_{\mathsf{mmse}} = \mathbb{E}[\mathbf{Z}|\mathbf{Z}_L] = \int \mathbf{Z} \, p(\mathbf{Z}|\mathbf{Z}_L) \, \mathrm{d}\mathbf{Z} \qquad (11)$$

## 3.2 Algorithm Details

The ML-Mat-VAMP for approximately computing the MAP and MMSE estimates is similar to the ML-VAMP method in [12, 33]. The specific iterations of ML-Mat-VAMP algorithm are shown in Algorithm 1. The algorithm produces estimates by a sequence of forward and backward pass updates denoted by superscripts $^+$ and $^-$ respectively. The estimates $\widehat{\mathbf{Z}}_\ell^\pm$ are constructed by solving sequential problems $\mathbf{Z} = \{\mathbf{Z}_\ell\}_{\ell=0}^{L-1}$ into a sequence of smaller problems each involving estimation of a single activation or preactivation $\mathbf{Z}_\ell$ via *estimation functions* $\{\mathbf{G}_\ell^\pm(\cdot)\}_{\ell=1}^{L-1}$ which are selected depending on whether one is interested in MAP or MMSE estimation.

To describe the estimation functions, we use the notation that, for a positive definite matrix $\mathbf{\Gamma}$, define the inner product $\langle \mathbf{A}, \mathbf{B} \rangle_\Gamma := \mathrm{Tr}(\mathbf{A}^\mathsf{T} \mathbf{B} \mathbf{\Gamma})$ and let $\|\mathbf{A}\|_\mathbf{\Gamma}$ denote the norm induced by this inner

---

**Algorithm 1** Multilayer Matrix VAMP (ML-Mat-VAMP)

---

**Require:** Estimators $\mathbf{G}_0^+, \mathbf{G}_L^-, \{\mathbf{G}_\ell^\pm\}_{\ell=1}^{L-1}$.

1: Set $\mathbf{R}_{0\ell}^- = \mathbf{0} \in \mathbb{R}^{n_\ell \times d}$ and initialize $\{\boldsymbol{\Gamma}_{0\ell}^-\}_{\ell=0}^{L-1} \in \mathbb{R}_{\succ 0}^{d \times d}$.

2: **for** $k = 0, 1, \ldots, N_{\text{it}} - 1$

3:     // Forward Pass

4:     $\widehat{\mathbf{Z}}_{k0}^+ = \mathbf{G}_0^+(\mathbf{R}_{k0}^-, \boldsymbol{\Gamma}_{k0}^-)$

5:     $\boldsymbol{\Lambda}_{k0}^+ = \left\langle \frac{\partial \mathbf{G}_0^+}{\partial \mathbf{R}_0^-}(\mathbf{R}_{k0}^-, \boldsymbol{\Gamma}_{k0}^-) \right\rangle^{-1} \boldsymbol{\Gamma}_{k,0}^-,$

6:     $\boldsymbol{\Gamma}_{k,0}^+ = \boldsymbol{\Lambda}_{k,0}^+ - \boldsymbol{\Gamma}_{k,0}^-$

7:     $\mathbf{R}_{k,0}^+ = (\widehat{\mathbf{Z}}_{k,0}^+ \boldsymbol{\Lambda}_{k,0}^+ - \mathbf{R}_{k,0}^- \boldsymbol{\Gamma}_{k,0}^-)(\boldsymbol{\Gamma}_{k,0}^+)^{-1}$

8:     **for** $\ell = 1, \ldots, L-1$ **do**

9:       $\widehat{\mathbf{Z}}_{k\ell}^+ = \mathbf{G}_\ell^+(\mathbf{R}_{k\ell}^-, \mathbf{R}_{k,\ell-1}^+, \boldsymbol{\Gamma}_{k\ell}^-, \boldsymbol{\Gamma}_{k,\ell-1}^+)$

10:      $\boldsymbol{\Lambda}_{k\ell}^+ = \left\langle \frac{\partial \mathbf{G}_\ell^+}{\partial \mathbf{R}_\ell^-}(\ldots) \right\rangle^{-1} \boldsymbol{\Gamma}_{k\ell}^-,$

11:      $\boldsymbol{\Gamma}_{k\ell}^+ = \boldsymbol{\Lambda}_{k\ell}^+ - \boldsymbol{\Gamma}_{k\ell}^-$

12:      $\mathbf{R}_{k\ell}^+ = (\widehat{\mathbf{Z}}_{k\ell}^+ \boldsymbol{\Lambda}_{k\ell}^+ - \mathbf{R}_{k\ell}^- \boldsymbol{\Gamma}_{k\ell}^-)(\boldsymbol{\Gamma}_{k\ell}^+)^{-1}$

13:     **end for**

14: // Backward Pass

15: $\widehat{\mathbf{Z}}_{k,L-1}^- = \mathbf{G}_L^-(\mathbf{R}_{k,L-1}^+, \boldsymbol{\Gamma}_{k,L-1}^+)$

16: $\boldsymbol{\Lambda}_{k,L-1}^- = \left\langle \frac{\partial \mathbf{G}_L^-}{\partial \mathbf{R}_{L-1}^+}(\mathbf{R}_{k,L-1}^+, \boldsymbol{\Gamma}_{k,L-1}^+) \right\rangle^{-1} \boldsymbol{\Gamma}_{k,L-1}^+,$

17: $\boldsymbol{\Gamma}_{k,L-1}^- = \boldsymbol{\Lambda}_{k,L-1}^- - \boldsymbol{\Gamma}_{k,L-1}^+$

18: $\mathbf{R}_{k+1,L-1}^- = (\widehat{\mathbf{Z}}_{k,L-1}^- \boldsymbol{\Lambda}_{k,L-1}^- - \mathbf{R}_{k,0}^+ \boldsymbol{\Gamma}_{k,0}^+)(\boldsymbol{\Gamma}_{k,0}^-)^{-1}$

19: **for** $\ell = L-1, \ldots, 1$ **do**

20:     $\widehat{\mathbf{Z}}_{k+1,\ell-1}^- = \mathbf{G}_\ell^-(\mathbf{R}_{k+1,\ell}^-, \mathbf{R}_{k,\ell-1}^+, \boldsymbol{\Gamma}_{k+1,\ell}^-, \boldsymbol{\Gamma}_{k,\ell-1}^+)$

21:     $\boldsymbol{\Lambda}_{k+1,\ell-1}^- = \left\langle \frac{\partial \mathbf{G}_\ell^-}{\partial \mathbf{R}_{\ell-1}^+}(\cdots) \right\rangle^{-1} \boldsymbol{\Gamma}_{k,\ell-1}^+,$

22:     $\boldsymbol{\Gamma}_{k+1,\ell}^- = \boldsymbol{\Lambda}_{k\ell}^- - \boldsymbol{\Gamma}_{k\ell}^+$

23:     $\mathbf{R}_{k+1,\ell-1}^- = (\widehat{\mathbf{Z}}_{k\ell}^- \boldsymbol{\Lambda}_{k\ell}^- - \mathbf{R}_{k\ell}^+ \boldsymbol{\Gamma}_{k\ell}^+)(\boldsymbol{\Gamma}_{k+1,\ell}^-)^{-1}$

24: **end for**

25: **end for**

---

product. For $\ell = 1, \ldots, L-1$ define the approximate belief functions

$$b_\ell(\mathbf{Z}_\ell, \mathbf{Z}_{\ell-1} | \mathbf{R}_\ell^-, \mathbf{R}_{\ell-1}^+, \boldsymbol{\Gamma}_\ell^-, \boldsymbol{\Gamma}_{\ell-1}^+) \propto p(\mathbf{Z}_\ell | \mathbf{Z}_{\ell-1}) e^{-\frac{1}{2}\|\mathbf{Z}_\ell - \mathbf{R}_\ell^-\|_{\boldsymbol{\Gamma}_\ell^-}^2 - \frac{1}{2}\|\mathbf{Z}_{\ell-1} - \mathbf{R}_{\ell-1}^+\|_{\boldsymbol{\Gamma}_{\ell-1}^+}^2}, \qquad (12)$$

where $\mathbf{Z}_\ell, \mathbf{R}_\ell^\pm \in \mathbb{R}^{n_\ell \times d}$ and $\boldsymbol{\Gamma}_\ell^\pm \in \mathbb{R}^{d \times d}$ for all $\ell = 0, 1, \ldots L$. Define $b_0(\mathbf{Z}_0 | \mathbf{R}_0^-, \boldsymbol{\Gamma}_0^-)$ and $b_L(\mathbf{Z}_{L-1} | \mathbf{R}_{L-1}^+, \boldsymbol{\Gamma}_{L-1}^+)$ similarly. The MAP and MMSE estimation functions are then given by the MAP and MMSE estimates for these belief densities,

$$\mathbf{G}_{\ell,\text{map}}^\pm = (\widehat{\mathbf{Z}}_\ell^+, \widehat{\mathbf{Z}}_{\ell-1}^-) = \text{argmax } b_\ell(\mathbf{Z}_\ell, \mathbf{Z}_{\ell-1}) \qquad \mathbf{G}_{\ell,\text{mmse}}^\pm = (\widehat{\mathbf{Z}}_\ell^+, \widehat{\mathbf{Z}}_{\ell-1}^-) = \mathbb{E}[(\mathbf{Z}_\ell, \mathbf{Z}_{\ell-1}) | b_\ell] \quad (13)$$

where the expectation is with respect to the normalized density proportional to $b_\ell$. Thus, the ML-Mat-VAMP algorithm reduces the joint estimation of the vectors $(\mathbf{Z}_0, \ldots, \mathbf{Z}_{L-1})$ to a sequence of simpler estimations on sub-problems with terms $(\mathbf{Z}_{\ell-1}, \mathbf{Z}_\ell)$. We refer to these subproblems as denoisers and denote their solutions by $\mathbf{G}_\ell^\pm$, so that $\widehat{\mathbf{Z}}_\ell^+ = \mathbf{G}_\ell^+$ and $\widehat{\mathbf{Z}}_{\ell-1}^- = \mathbf{G}_\ell^-$ corresponding to lines 9 and 20 of Algorithm 1. The denoisers $\mathbf{G}_0^+$ and $\mathbf{G}_L^-$, which provide updates to $\widehat{\mathbf{Z}}_0^+$ and $\widehat{\mathbf{Z}}_{L-1}^-$, are defined in a similar manner via $b_0$ and $b_L$ respectively.

The estimation functions (13) can be easily computed for the multi-layer matrix network. An important characteristic of these estimators is that they can be computed using maps which are row-wise separable over their inputs and hence are easily parallelizable. To simplify notation, we denote the precision parameters for denoisers $\mathbf{G}_\ell^\pm$ in the $k^{\text{th}}$ iteration by

$$\boldsymbol{\Theta}_{k\ell}^+ := (\boldsymbol{\Gamma}_{k\ell}^-, \boldsymbol{\Gamma}_{k,\ell-1}^+), \quad \boldsymbol{\Theta}_{k\ell}^- := (\boldsymbol{\Gamma}_{k+1,\ell}^-, \boldsymbol{\Gamma}_{k,\ell-1}^+), \qquad \boldsymbol{\Theta}_{k0}^+ := \boldsymbol{\Gamma}_{k0}^-, \qquad \boldsymbol{\Theta}_{kL}^- := \boldsymbol{\Gamma}_{k,L-1}^+. \quad (14)$$

**Non-linear layers:** For $\ell$ even, since the rows of $\boldsymbol{\Xi}_\ell$ are i.i.d., the belief density $b_\ell(\mathbf{Z}_\ell, \mathbf{Z}_{\ell-1} | \cdot)$ from (12) factors as a product across rows, $b_\ell(\mathbf{Z}_\ell, \mathbf{Z}_{\ell-1}) = \prod_n b_\ell([\mathbf{Z}_\ell]_{n:}, [\mathbf{Z}_{\ell-1}]_{n:})$. Thus, the MAP and MMSE estimates (13) can be performed over $d$-dimensional variables where $d$ is the number of entries in each row. There is no joint estimation across the different $n_\ell$ rows.

**Linear layers:** When $\ell$ is odd, the density $b_\ell(\mathbf{Z}_\ell, \mathbf{Z}_{\ell-1} | \cdot)$ in (12) is a Gaussian. Hence, the MAP and MMSE estimates agree and can be computed via least squares. Although for linear layers $[\mathbf{G}_\ell^+, \mathbf{G}_\ell^-](\mathbf{R}_\ell^-, \mathbf{R}_{\ell-1}^+, \boldsymbol{\Theta}_\ell)$ is not row-wise separable over $(\mathbf{R}_\ell^-, \mathbf{R}_{\ell-1})$, it can be computed using another row-wise denoiser $[\widetilde{\mathbf{G}}_\ell^+, \widetilde{\mathbf{G}}_\ell^-]$ via the SVD of the weight matrix $\mathbf{W}_\ell = \mathbf{V}_\ell \text{diag}(\mathbf{S}_\ell) \mathbf{V}_{\ell-1}$ as

follows. Note that the SVD is only needed to be performed once.:

$$[\mathbf{G}_\ell^+, \mathbf{G}_\ell^-](\mathbf{R}_\ell, \mathbf{R}_{\ell-1}, \boldsymbol{\Theta}_\ell) = \operatorname*{argmax}_{\mathbf{Z}_\ell, \mathbf{Z}_{\ell-1}} \|\mathbf{Z}_\ell - \mathbf{W}_\ell \mathbf{Z}_{\ell-1} - \mathbf{B}_\ell\|_{\mathbf{N}_\ell}^2 + \|\mathbf{Z}_\ell - \mathbf{R}_\ell^-\|_{\boldsymbol{\Gamma}_\ell^-}^2 + \|\mathbf{Z}_{\ell-1} - \mathbf{R}_{\ell-1}^+\|_{\boldsymbol{\Gamma}_{\ell-1}^+}^2$$

$$\overset{(a)}{=} \operatorname*{argmax}_{\mathbf{Z}_\ell, \mathbf{Z}_{\ell-1}} \|\mathbf{V}_\ell^\top \mathbf{Z}_\ell - \operatorname{diag}(\mathbf{S}_\ell)\mathbf{V}_{\ell-1}\mathbf{Z}_{\ell-1} - \mathbf{V}_\ell^\top \mathbf{B}_\ell\|_{\mathbf{N}_\ell}^2 + \|\mathbf{V}_\ell^\top \mathbf{Z}_\ell - \mathbf{V}_\ell^\top \mathbf{R}_\ell^-\|_{\boldsymbol{\Gamma}_\ell^-}^2 + \|\mathbf{V}_{\ell-1}\mathbf{Z}_{\ell-1} - \mathbf{V}_{\ell-1}\mathbf{R}_{\ell-1}^+\|_{\boldsymbol{\Gamma}_{\ell-1}^+}^2$$

$$\overset{(b)}{=} [\mathbf{V}_\ell^\top \widetilde{\mathbf{G}}_\ell^+, \mathbf{V}_{\ell-1}\widetilde{\mathbf{G}}_\ell^-](\mathbf{V}_\ell^\top \mathbf{R}_\ell, \mathbf{V}_{\ell-1}\mathbf{R}_{\ell-1}, \boldsymbol{\Theta}_\ell)$$

where (a) follows from the rotational invariance of the norm, and (b) follows from the definition of denoiser $[\widetilde{\mathbf{G}}_\ell^+, \widetilde{\mathbf{G}}_\ell^-](\widetilde{\mathbf{R}}_\ell^-, \widetilde{\mathbf{R}}_{\ell-1}^+, \boldsymbol{\Theta}_\ell)$ given below

$$[\widetilde{\mathbf{G}}_\ell^+, \widetilde{\mathbf{G}}_\ell^-] := \operatorname*{argmax}_{\widetilde{\mathbf{Z}}_\ell, \widetilde{\mathbf{Z}}_{\ell-1}} \left\|\widetilde{\mathbf{Z}}_\ell - \operatorname{diag}(\mathbf{S}_\ell)\widetilde{\mathbf{Z}}_{\ell-1} - \widetilde{\mathbf{B}}_\ell\right\|_{\mathbf{N}_\ell}^2 \left\|\widetilde{\mathbf{Z}}_\ell - \widetilde{\mathbf{R}}_\ell^-\right\|_{\boldsymbol{\Gamma}_\ell^-}^2 + \left\|\widetilde{\mathbf{Z}}_{\ell-1} - \widetilde{\mathbf{R}}_{\ell-1}^+\right\|_{\boldsymbol{\Gamma}_{\ell-1}^+}^2 \quad (15)$$

Note that the optimization problem in (15), is decomposable accross the rows of variables $\widetilde{\mathbf{Z}}_\ell$ and $\widetilde{\mathbf{Z}}_{\ell-1}$, and hence $[\widetilde{\mathbf{G}}_\ell^+, \widetilde{\mathbf{G}}_\ell^-]$ operates row-wise on its inputs.

**Fixed Points:** We note that the fixed points of the ML-Mat-VAMP algorithm can be shown to be KKT points of the variational formulations of (11), omitted here due to lack of space. This is a direct extention of results from Section 3 of [35]. In particular, we can show that the ML-Mat-VAMP in the MAP inference case is a preconditioned *Peaceman-Rachford splitting* ADMM type algorithm [40].

## 4   Analysis in the Large System Limit

We follow the analysis framework of the ML-VAMP work [12, 33], which is itself based on the original AMP analysis in [3]. This analysis is based on considering the asymptotics of certain large random problem instances. We essentially show that under certain assumptions, as the dimension goes to infinity the behavior of the ML-Mat-VAMP algorithm can be characterized by a set of equations that describe how the distribution of rows of hidden matrices evolve at each iteration of the algorithm for all the layers. Specifically, we consider a sequence of problems (1) indexed by $N$ such that for each problem the dimensions $n_\ell(N)$ are growing so that $\lim_{N\to\infty}\frac{n_\ell}{N} = \beta_\ell \in (0,\infty)$ are scalar constants. Note that $d$ is finite and does not grow with $N$.

**Distributions of weight matrices:** For $\ell = 1, 3, \ldots, L-1$, we assume that the weight matrices $\mathbf{W}_\ell$ are generated via the singular value decomposition, $\mathbf{W}_\ell = \mathbf{V}_\ell \operatorname{diag}(\mathbf{S}_\ell)\mathbf{V}_{\ell-1}$ where $\mathbf{V}_\ell \in \mathbb{R}^{n_\ell \times n_\ell}$ are Haar distributed over orthonormal matrices and $\mathbf{S}_\ell = (s_{\ell,1}, \ldots, s_{\ell,\min\{n_\ell,n_{\ell-1}\}})$. We will describe the distribution of the components $\mathbf{S}_\ell$ momentarily.

**Assumption on Denoisers:** We assume that the non-linear denoisers $\mathbf{G}_{2k}^\pm$ act row-wise on their inputs $(\mathbf{R}_{2k}^-, \mathbf{R}_{2k-1}^+)$. Further these operators and their Jacobian matrices $\frac{\partial \mathbf{G}_{2k}^+}{\partial \mathbf{R}_{2k}}, \frac{\partial \mathbf{G}_{2k}^-}{\partial \mathbf{R}_{2k-1}^+}, \frac{\partial \mathbf{G}_0^+}{\partial \mathbf{R}_0^-}, \frac{\partial \mathbf{G}_L^-}{\partial \mathbf{R}_{L-1}^+}$ are *uniformly Lipschitz continuous*, the definition of which is provided in Appendix B.

**Assumption on initialization, true variables:** The distribution of the remaining variables is described by a weak limit: For a matrix sequence $\boldsymbol{X} := \boldsymbol{X}(N) \in \mathbb{R}^{N \times d}$, by the notation $\boldsymbol{X} \overset{2}{\Rightarrow} X$ we mean that there exists a random variable $X$ in $\mathbb{R}^d$ with $\mathbb{E}\|X\|^2 < \infty$ such that $\lim_{N\to\infty} \frac{1}{N}\sum_{i=1}^N \psi(\boldsymbol{X}_{i:}) = \mathbb{E}\,\psi(X)$ almost surely, for any bounded continuous function $\psi : \mathbb{R}^d \to \mathbb{R}$, as well as for quadratic functions $\boldsymbol{x}^\top \boldsymbol{P}\boldsymbol{x}$ for any $\boldsymbol{P} \in \mathbb{R}_{\succeq 0}^{d \times d}$. This is also referred to as Wasserstein-2 convergence [30]. For e.g., this property is satisfied for a random $\boldsymbol{X}$ with i.i.d. rows with bounded second moments, but is more general, since it applies to deterministic matrix sequences as well. More details on this weak limit are given in Appendix B.

Let $\overline{\mathbf{B}}_\ell := \mathbf{V}_\ell^\top \mathbf{B}_\ell$, and $\overline{\mathbf{S}}_\ell \in \mathbb{R}^{n_\ell}$ be the zero-padded vector of singular values of $\mathbf{W}_\ell$, and let $\boldsymbol{\tau}_{0\ell}^- \in \mathbb{R}_{\succ 0}^{d \times d}$. Then we assume that the following empirical convergences hold.$(\boldsymbol{\Xi}_\ell, \mathbf{R}_{0\ell}^- - \mathbf{Z}_\ell^0) \overset{2}{\Rightarrow} (\Xi_\ell, Q_{0\ell}^-)$ for even $\ell$ and $(\overline{\mathbf{S}}_\ell, \overline{\mathbf{B}}_\ell, \boldsymbol{\Xi}_\ell, \mathbf{V}_\ell^\top (\mathbf{R}_{0\ell}^- - \mathbf{Z}_\ell^0)) \overset{2}{\Rightarrow} (S_\ell, \overline{B}_\ell, \Xi_\ell, Q_{0\ell}^-)$, for odd $\ell$. Here $S_\ell \in \mathbb{R}_{\geq 0}$ is bounded, $\overline{B}_\ell \in \mathbb{R}^d$ is bounded, $\Xi_{2\ell-1} \sim \mathcal{N}(\mathbf{0}, \mathbf{N}_{2\ell-1}^{-1})$, and $Q_{0\ell}^- \sim \mathcal{N}(\mathbf{0}, \overline{\Gamma}_{0\ell}^-)$, for $\ell = 0, 1, \ldots, L-1$

are all pairwise independent random variables. Additionally, we assume that $\mathbf{Z}_0^0 \overset{2}{\Rightarrow} Z^0$ and that the sequence of initial matrices $\{\boldsymbol{\Gamma}_{0\ell}^-\}$ satisfies the following pointwise convergence

$$\boldsymbol{\Gamma}_{0\ell}^-(N) \to \overline{\boldsymbol{\Gamma}}_{0\ell}^-, \quad \ell = 0, 1, \ldots, L-1 \tag{16}$$

## 4.1 Main Result

The main result of this paper concerns the empirical distribution of the rows $[\widehat{\mathbf{Z}}_\ell^\pm]_{n:}, [\mathbf{R}_\ell^\pm]_{n:}$ of the iterates of Algorithm 1. It characterizes the asymptotic behaviour of these empirical distributions in terms of $d$-dimensional random vectors which are either Gaussians or functions of Gaussians. Let $G_\ell^\pm$ denote maps $\mathbb{R}^{1\times d} \to \mathbb{R}^{1\times d}$, such that (13), i.e., $[\mathbf{G}_\ell^\pm(\mathbf{R}_\ell^-, \mathbf{R}_{\ell-1}^+, \boldsymbol{\Theta})]_{n:} = G_\ell^\pm([\mathbf{R}_\ell^-]_{n:}, [\mathbf{R}_{\ell-1}^+]_{n:}, \boldsymbol{\Theta})$. Having stated the requisite definitions and assumptions, we can now state our main result.

**Theorem 1.** *For a fixed iteration index $k \geq 0$, there exist deterministic matrices $\mathbf{K}_{k\ell}^+ \in \mathbb{R}_{\succ 0}^{2d\times 2d}$, and $\boldsymbol{\tau}_{k\ell}^-, \overline{\boldsymbol{\Gamma}}_{k\ell}^+$ and $\overline{\boldsymbol{\Gamma}}_{k\ell}^-, \in \mathbb{R}_{\succ 0}^{d\times d}$ such that for even $\ell$:*

$$\left(\mathbf{Z}_{\ell-1}^0, \mathbf{Z}_\ell^0, \widehat{\mathbf{Z}}_{k,\ell-1}^-, \widehat{\mathbf{Z}}_{k\ell}^+\right) \overset{2}{\Rightarrow} \left(\mathsf{A}, \widetilde{\mathsf{A}}, G_\ell^-(\mathsf{C} + \widetilde{\mathsf{A}}, \mathsf{B} + \mathsf{A}, \overline{\boldsymbol{\Gamma}}_{k\ell}^-, \overline{\boldsymbol{\Gamma}}_{k,\ell-1}^+), G_\ell^+(\mathsf{C} + \widetilde{\mathsf{A}}, \mathsf{B} + \mathsf{A}, \overline{\boldsymbol{\Gamma}}_{k\ell}^-, \overline{\boldsymbol{\Gamma}}_{k,\ell-1}^+)\right)$$

*where $(\mathsf{A}, \mathsf{B}) \sim \mathcal{N}(\mathbf{0}, \mathbf{K}_{k,\ell-1}^+)$, $\mathsf{C} \sim \mathcal{N}(\mathbf{0}, \boldsymbol{\tau}_{k\ell}^-)$, $\widetilde{\mathsf{A}} = \phi_\ell(\mathsf{A}, \Xi_\ell)$ and $(\mathsf{A}, \mathsf{B}), \mathsf{C}$ are independent. For $\ell = 0$, the same result holds where the 1st and 3rd terms are dropped, whereas for $\ell = L$, the 2nd and 4th terms are dropped. Similarly, for odd $\ell$:*

$$\left(\mathbf{V}_{\ell-1}^\mathsf{T}\mathbf{Z}_{\ell-1}^0, \mathbf{V}_{\ell-1}^\mathsf{T}\mathbf{Z}_\ell^0, \mathbf{V}_\ell\widehat{\mathbf{Z}}_{k,\ell-1}^-, \mathbf{V}_\ell\widehat{\mathbf{Z}}_{k\ell}^+\right) \overset{2}{\Rightarrow}$$

$$\left(\mathsf{A}, \widetilde{\mathsf{A}}, G_\ell^-(\mathsf{C} + \widetilde{\mathsf{A}}, \mathsf{B} + \mathsf{A}, \overline{\boldsymbol{\Gamma}}_{k\ell}^-, \overline{\boldsymbol{\Gamma}}_{k,\ell-1}^+), G_\ell^+(\mathsf{C} + \widetilde{\mathsf{A}}, \mathsf{B} + \mathsf{A}, \overline{\boldsymbol{\Gamma}}_{k\ell}^-, \overline{\boldsymbol{\Gamma}}_{k,\ell-1}^+)\right)$$

*where $(\mathsf{A}, \mathsf{B}) \sim \mathcal{N}(\mathbf{0}, \mathbf{K}_{k,\ell-1}^+)$, $\mathsf{C} \sim \mathcal{N}(\mathbf{0}, \boldsymbol{\tau}_{k\ell}^-)$, $\widetilde{\mathsf{A}} = S_\ell \mathsf{A} + \overline{B}_\ell + \Xi_\ell$ and $(\mathsf{A}, \mathsf{B}), \mathsf{C}$ are independent.*

*Furthermore for $\ell = 0, 1, \ldots L-1$, we have*

$$(\boldsymbol{\Gamma}_{k\ell}^\pm, \boldsymbol{\Lambda}_{k\ell}^\pm) \xrightarrow{a.s.} (\overline{\boldsymbol{\Gamma}}_{k\ell}^\pm, \overline{\boldsymbol{\Lambda}}_{k\ell}^\pm).$$

The parameters in the distribution, $\{\mathbf{K}_{k\ell}^+, \boldsymbol{\tau}_{k\ell}^-, \overline{\boldsymbol{\Gamma}}_{k\ell}^\pm, \overline{\boldsymbol{\Lambda}}_{k\ell}^\pm\}$ are deterministic and can be computed via a set of recursive equations called the *state evolution* or SE. The SE equations are provided in Appendix A The result is similar to those for ML-VAMP in [12, 35] except that the SE equations for ML-Mat-VAMP involve $d \times d$ and $2d \times 2d$ matrix terms; the ML-VAMP SE only requires scalar and $2 \times 2$ matrix terms. The result holds for both MAP inference and MMSE inference, the only difference is implicit, i.e., the choice of denoiser $\mathbf{G}_\ell(\cdot)$ from eqn. (13).

The importance of Theorem 1 is that the rows of the iterates of the ML-Mat-VAMP Algorithm ($\widehat{\mathbf{Z}}_{k,\ell-1}^-, \widehat{\mathbf{Z}}_{k\ell}^+$ in Algorithm 1) and the rows of the corresponding true values, $\mathbf{Z}_{\ell-1}^0, \mathbf{Z}_\ell^0$, have a simple, asymptotic random vector description of a typical row. We will call this the "row-wise" model. According to this model, for even $\ell$, the rows of $\mathbf{Z}_{\ell-1}^0$ converge to a Gaussian $\mathsf{A} \in \mathbb{R}^d$ and the rows of $\mathbf{Z}_\ell^0$ converge to the output of the Gaussian through the row-wise function $\phi_\ell$, $\widetilde{\mathsf{A}} = \phi_\ell(\mathsf{A}, \Xi_\ell)$. Then the rows of the estimates $\widehat{\mathbf{Z}}_{k,\ell-1}^-, \widehat{\mathbf{Z}}_{k\ell}^+$ asymptotically approach the outputs of row-wise estimation function $G^-(\cdot)$ and $G^+(\cdot)$ supplied by $\mathsf{A}$ and $\widetilde{\mathsf{A}}$ corrupted with Gaussian noise. A similar convergence holds for odd $\ell$.

This "row-wise" model enables exact an analysis of the performance of the estimates at each iteration. For example, to compute a weighted mean squared error (MSE) metric at iteration $k$, the convergence shows that,

$$\frac{1}{n_\ell}\left\|\widehat{\mathbf{Z}}_{k\ell}^+ - \mathbf{Z}_\ell^0\right\|_{\mathbf{H}}^2 \xrightarrow{a.s.} \mathbb{E}\|\mathbf{G}_\ell^+(\mathsf{C} + \widetilde{\mathsf{A}}, \mathsf{B} + \mathsf{A}, \boldsymbol{\Theta}_{k\ell}) - \widetilde{\mathsf{A}}\|_{\mathbf{H}}^2,$$

for even $\ell$ and any positive semi-definite matrix $\mathbf{H} \in \mathbb{R}^{d\times d}$. The norm on the left-hand above acts row-wise, $\|\mathbf{Z}\|_{\mathbf{H}}^2 := \sum_i \|\mathbf{Z}_{i:}\|_{\mathbf{H}}^2$. Hence, this asymptotic MSE can be evaluated via expectations of $d$-dimensional variables from the SE. Similarly, one can obtain exact answers for any other row-wise performance metric of $\{(\widehat{\mathbf{Z}}_{k\ell}^\pm, \mathbf{Z}_\ell^0)\}_\ell$ for any $k$.

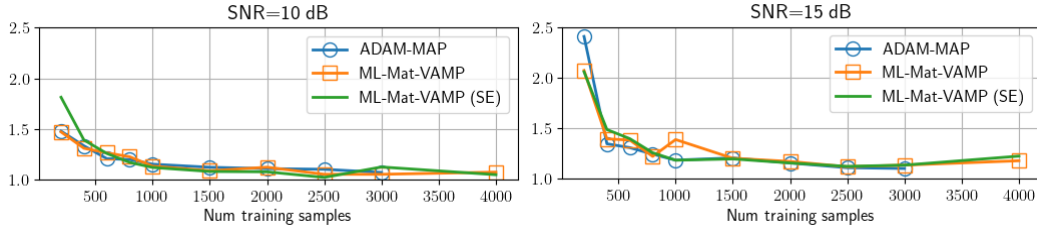

Figure 2: Test error in learning the first layer of a 2 layer neural network using ADAM-based gradient descent, ML-Mat-VAMP and its state evolution prediction.

## 5  Numerical Experiments

We consider the problem of learning the input layer of a two layer neural network as described in Section 2.3. We learn the weights $\boldsymbol{F}_1$ of the first layer of a two-layer network by solving problem (9). The large system limit analysis in this case corresponds to the input size $n_{\text{in}}$ and number of samples $N$ going to infinity with the number of hidden units being fixed. Our experiment take $d = 4$ hidden units, $N_{\text{in}} = 100$ input units, $N_{\text{out}} = 1$ output unit, sigmoid activations and variable number of samples $N$. The weight vectors $\boldsymbol{F}_1$ and $\boldsymbol{F}_2$ are generated as i.i.d. Gaussians with zero mean and unit variance. The input $\mathbf{X}$ is also i.i.d. Gaussians with variance $1/N_{\text{in}}$ so that the average pre-activation has unit variance. Output noise is added at two levels of 10 and 15 dB relative to the mean. We generate 1000 test samples and a variable number of training samples that ranges from 200 to 4000. For each trial and number of training samples, we compare three methods: (i) MAP estimation where the MAP loss function is minimized by the ADAM optimizer [21] in the Keras package of Tensorflow; (ii) Algorithm 1 run for 20 iterations and (iii) the state evolution prediction. The ADAM algorithm is run for 100 epochs with a learning rate = 0.01. The expectations in the SE are estimated via Monte-Carlo sampling (hence there is some variation).

Given an estimate $\widehat{\mathbf{F}}_1$ and true value $\mathbf{F}_1^0$, we can compute the test error as follows: Given a new sample $\mathbf{x}$, the true and predicted pre-activations will be $\mathbf{z}_1 = (\mathbf{F}_1^0)^{\mathsf{T}} \mathbf{x}$ and $\widehat{\mathbf{z}}_1 = \widehat{\mathbf{F}}_1^{\mathsf{T}} \mathbf{x}$. Thus, if the new sample $\mathbf{x} \sim \mathcal{N}(0, \frac{1}{N_{\text{in}}} \mathbf{I})$, the true and predicted pre-activations, $(\mathbf{z}_1, \widehat{\mathbf{z}}_1)$, will be jointly Gaussian with covariance equal to the empirical $2d \times 2d$ covariance matrix of the rows of $\mathbf{F}_1^0$ and $\widehat{\mathbf{F}}_1$:

$$\mathbf{K} := \tfrac{1}{N_{\text{in}}} \sum_{k=1}^{N_{\text{in}}} \mathbf{u}_k^{\mathsf{T}} \mathbf{u}_k, \quad \mathbf{u}_k = \left[ \mathbf{F}_{1,k:} \; \widehat{\mathbf{F}}_{1,k:} \right] \tag{17}$$

From this covariance matrix, we can estimate the test error, $\mathbb{E}|y - \widehat{y}|^2 = \mathbb{E}|\mathbf{F}_2^{\mathsf{T}}(\sigma(\mathbf{z}_1) - \sigma(\widehat{\mathbf{z}}_1)|^2$, where the expectation is taken over the Gaussian $(\mathbf{z}_1, \widehat{\mathbf{z}}_1)$ with covariance $\mathbf{K}$. Also, since (17) is a row-wise operation, it can be predicted from the ML-Mat-VAMP SE. Thus, the SE can also predict the asymptotic test error. The normalized test error for ADAM-MAP, ML-Mat-VAMP and the ML-Mat-VAMP SE are plotted in Fig. 2. The normalized test error is defined as the ratio of the MSE on the test samples to the optimal MSE. Hence, a normalized MSE of one is the minimum value.

Note that since ADAM and ML-Mat-VAMP are solving the same optimization problem, they perform similarly as expected. The main message of this paper is not to develop an algorithm that outperforms ADAM, but rather an algorithm that has theoretical guarantees. The key property of ML-Mat-VAMP is that its asymptotic behavior at all the iterations can be exactly predicted by the state evolution equations. In this example, Fig. 2 shows that the normalized test MSE predicted via state evolution (plotted in green) matches the normalized MSE of ML-Mat-VAMP estimates (plotted in orange).

## 6  Conclusions

We have developed a general framework for analyzing inference in multi-layer networks with matrix valued quantities in certain high-dimensional random settings. For learning the input layer of a two layer network, the methods enables precise predictions of the expected test error as a function of the parameter statistics, numbers of samples and noise level. This analysis can be valuable in understanding key properties such as generalization error, for example using ML-VAMP, Emami et al. [11] characterizes the generalization error of GLMs under a variety of feature distributions and train-test mismatch. Future work will look to extend these to more complex networks.

## Broader Impact

In statistical physics, systems with a large number of degrees of freedom often admit a simplified macroscopic description. Modern neural networks have thousands of hidden units and billions of free parameters; is there an analogous macroscopic description of the dynamics of multi-layer neural networks? This paper identifies some of these macroscopic descriptions that can be used to analyze a large class of optimization problems (See Section 2 for examples) arising in Signal Processing, Data Science, and Machine Learning. The techniques developped in this paper allow analyzing and understanding the fundamental limits of learning in 1 and 2 layer neural networks which are basic building blocks in modern machine learning pipelines.

## Acknowledgements and Funding Disclosure

The work of P. Schniter was supported by NSF grant 1716388. The work of P. Pandit, M. Saharee-Ardakan and A. K. Fletcher was supported in part by the NSF Grants 1738285 and 1738286, ONR Grant N00014-15-1-2677. The work of S. Rangan was supported in part by NSF grants 1116589, 1302336, and 1547332, NIST, SRC and the the industrial affiliates of NYU Wireless.

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
