[Supplementary Material]

# Supplementary materials: "Matrix Inference and Estimation in Multi-Layer Models"

## Appendix A    State Evolution Equations

The state evolution equations given in Algo. 2 define an iteration indexed by $k$ of constant matrices $\{\mathbf{K}_{k\ell}^+, \boldsymbol{\tau}_{kl}^-, \overline{\boldsymbol{\Gamma}}_{kl}^\pm\}_{\ell=0}^L$. These constants appear in the statement of the main result in Theorem 1. The iterations in Algo. 2 also iteratively define a few $\mathbb{R}^{1\times d}$ valued random vectors $\{Q_\ell^0, P_\ell^0, Q_{k\ell}^\pm, P_{k\ell}^\pm\}$ which are either multivariate Gaussian or functions of Multivariate Gaussians. In order to state Algorithm 2, we need to define certain random variables and functions appearing therein which are described below. Let $\mathcal{L}_{\text{odd}} = \{1, 3, \ldots, L-1\}$ and $\mathcal{L}_{\text{even}} = \{2, 4, \ldots, L-2\}$.

Define $\{\overline{\boldsymbol{\Theta}}_{k\ell}^\pm\}$ similar to $\boldsymbol{\Theta}_{k\ell}^\pm$ from equation (14) using $\{\overline{\boldsymbol{\Gamma}}_{k\ell}^\pm\}$. Further, for $\ell = 1, 2, \ldots, L-1$ define

$$\overline{\boldsymbol{\Omega}}_{k\ell}^+ := (\overline{\boldsymbol{\Lambda}}_{k\ell}^+, \overline{\boldsymbol{\Gamma}}_{k\ell}^+, \overline{\boldsymbol{\Gamma}}_{k\ell}^-), \ \overline{\boldsymbol{\Omega}}_{k\ell}^- := (\overline{\boldsymbol{\Lambda}}_{k,\ell-1}^+, \overline{\boldsymbol{\Gamma}}_{k,\ell-1}^-, \overline{\boldsymbol{\Gamma}}_{k,\ell-1}^-),$$

and $\overline{\boldsymbol{\Omega}}_{k0}^+$ and $\overline{\boldsymbol{\Omega}}_{kL}^-$. Now define random variables $W_\ell$ as

$$\begin{aligned}
&W_0 = Z_0^0, \ W_L = (Y, \Xi_L), \ W_\ell = \Xi_\ell, \quad \forall \ell \in \mathcal{L}_{\text{even}}, \\
&W_\ell = (S_\ell, \overline{B}_\ell, \Xi_\ell), \quad\quad\quad\quad\quad\quad \forall \ell \in \mathcal{L}_{\text{odd}}.
\end{aligned} \tag{18}$$

Define functions $\{f_\ell^0\}_{\ell=1}^L$ as

$$\begin{aligned}
&f_\ell^0(P_{\ell-1}^0, W_\ell) := S_\ell P_{\ell-1}^0 + \overline{B}_\ell + \Xi_\ell, \quad \forall \ell \in \mathcal{L}_{\text{odd}}, \\
&f_\ell^0(P_{\ell-1}^0, W_\ell) := \phi_\ell(P_{\ell-1}^0, \Xi_\ell), \quad \forall \ell \in \mathcal{L}_{\text{even}} \cup \{L\}.
\end{aligned} \tag{19}$$

and using (14) define functions $\{h_\ell^\pm,\}_{\ell=1}^L$, $h_0^+$ and $h_L^-$ as

$$\begin{aligned}
&h_\ell^\pm(P_{\ell-1}^0, P_{\ell-1}^+, Q_\ell^-, W_\ell, \boldsymbol{\Theta}_{k\ell}^\pm) = G_\ell^\pm(Q_\ell^- + Q_\ell^0, P_{\ell-1}^+ + P_{\ell-1}^0, \boldsymbol{\Theta}_{k\ell}^\pm), \ \forall \ell \in \mathcal{L}_{\text{even}}, \\
&h_\ell^\pm(P_{\ell-1}^0, P_{\ell-1}^+, Q_\ell^-, W_\ell, \boldsymbol{\Theta}_{k\ell}^\pm) = \widetilde{G}_\ell^\pm(Q_\ell^- + Q_\ell^0, P_{\ell-1}^+ + P_{\ell-1}^0, \boldsymbol{\Theta}_{k\ell}^\pm), \ \forall \ell \in \mathcal{L}_{\text{odd}} \\
&h_0^+(Q_0^-, W_0, \boldsymbol{\Theta}_{k0}^+) = G_0^+(Q_0^- + W_0, \boldsymbol{\Theta}_{k0}^+), \\
&h_L^-(P_{L-1}^0, P_{L-1}^+, W_L, \boldsymbol{\Theta}_{kL}^-) = G_L^-(P_{L-1}^+ + P_{L-1}^0, \boldsymbol{\Theta}_{kL}^-).
\end{aligned} \tag{20}$$

Note that $[G_\ell^+, G_\ell^-]$ and $[\widetilde{G}_\ell^+, \widetilde{G}_\ell^-]$ are maps from $\mathbb{R}^{1\times d} \to \mathbb{R}^{1\times d}$ such that their row-wise extensions are the denoisers $[\mathbf{G}_\ell^+, \mathbf{G}_\ell^-]$ and $[\widetilde{\mathbf{G}}_\ell^+, \widetilde{\mathbf{G}}_\ell^-]$ respectively. Using (20) define functions $\{f_\ell^\pm\}_{\ell=1}^{L-1}$, $f_0^+$ and $f_L^-$ as

$$\begin{aligned}
&f_\ell^+(P_{\ell-1}^0, P_{\ell-1}^+, Q_\ell^-, W_\ell, \boldsymbol{\Omega}_{k\ell}^+) = \left[\left(h_\ell^+ - Q_\ell^0\right)\boldsymbol{\Lambda}_{k\ell}^+ - Q_\ell^-\boldsymbol{\Gamma}_{k\ell}^-\right](\boldsymbol{\Gamma}_{k\ell}^+)^{-1}, \\
&f_\ell^-(P_{\ell-1}^0, P_{\ell-1}^+, Q_\ell^-, W_\ell, \boldsymbol{\Omega}_{k\ell}^-) = \left[\left(h_\ell^- - P_{\ell-1}^0\right)\boldsymbol{\Lambda}_{k,\ell-1}^- - P_{\ell-1}^+\boldsymbol{\Gamma}_{k,\ell-1}^+\right](\boldsymbol{\Gamma}_{k,\ell-1}^-)^{-1}. \\
&f_0^+(Q_0^-, W_0, \boldsymbol{\Omega}_{k0}^+) = \left[\left(h_0^+ - W_0\right)\boldsymbol{\Lambda}_{k0}^+ - Q_0^-\boldsymbol{\Gamma}_{k0}^-\right](\boldsymbol{\Gamma}_{k0}^+)^{-1}, \\
&f_L^-(P_{L-1}^0, P_{L-1}^+, W_L, \boldsymbol{\Omega}_{kL}^-) = \left[\left(h_L^- - P_{L-1}^0\right)\boldsymbol{\Lambda}_{k,L-1}^- - P_{L-1}^+\boldsymbol{\Gamma}_{k,L-1}^+\right](\boldsymbol{\Gamma}_{k,L-1}^-)^{-1}.
\end{aligned} \tag{21}$$

## Appendix B    Large System Limit Details

The analysis of Algorithm 1 in the large system limit is based on [3] and is by now standard in the theory of AMP-based algorithms. The goal is to characterize ensemble row-wise averages of iterates of the algorithm using *simpler* finite-dimensional random variables which are either Gaussians or functions of Gaussians. To that end, we start by defining some key terms needed in this analysis.

**Definition 1** (Pseudo-Lipschitz continuity). For a given $p \geq 1$, a map $\mathbf{g} : \mathbb{R}^{1\times d} \to \mathbb{R}^{1\times r}$ is called pseudo-Lipschitz of order $p$ if for any $\mathbf{r}_1, \mathbf{r}_2 \in \mathbb{R}^d$ we have,

$$\|\mathbf{g}(\mathbf{r}_1) - \mathbf{g}(\mathbf{r}_2)\| \leq C\|\mathbf{r}_1 - \mathbf{r}_2\|\left(1 + \|\mathbf{r}_1\|^{p-1} + \|\mathbf{r}_2\|^{p-1}\right)$$

**Definition 2** (Empirical convergence of rows of a matrix sequence). Consider a matrix-sequence $\{\mathbf{X}^{(N)}\}_{N=1}^\infty$ with $\mathbf{X}^{(N)} \in \mathbb{R}^{N\times d}$. For a finite $p \geq 1$, let $X \in (\mathbb{R}^d, \mathcal{R}^d)$ be a $\mathcal{R}^d$-measurable random variable with bounded moment $\mathbb{E}\|X\|_p^p < \infty$. We say the rows of matrix sequence $\{\mathbf{X}^{(N)}\}$ *converge empirically to $X$ with $p^{th}$ order moments* if for all pseudo-Lipschitz continuous functions $f(\cdot)$ of

**Algorithm 2** State Evolution for ML-Mat-VAMP (Algo. 1)

---

**Require:** Functions $\{f_\ell^0\}$ from (19), $\{h_\ell^\pm\}$ from (20), and $\{f_\ell^\pm\}$ from (21). Perturbation random variables $\{W_\ell\}$ from (18). Initial random vectors $\{Q_{0\ell}^-\}_{\ell=0}^{L-1}$ with Initial covariance matrices $\{\boldsymbol{\tau}_{0\ell}^-\}_{\ell=0}^{L-1}$ from Section 4. Initial matrices $\{\overline{\boldsymbol{\Gamma}}_{0\ell}^-\}_{\ell=0}^{L}$ from (16).

---

1: // Initial Pass
2: $Q_0^0 = W_0$, $\boldsymbol{\tau}_0^0 = \mathrm{Cov}(Q_0^0)$ and $P_0^0 \sim \mathcal{N}(\mathbf{0}, \boldsymbol{\tau}_0^0)$
3: **for** $\ell = 1, \ldots, L-1$ **do**
4: $\quad Q_\ell^0 = f_\ell^0(P_{\ell-1}^0, W_\ell)$
5: $\quad P_\ell^0 \sim \mathcal{N}(\mathbf{0}, \boldsymbol{\tau}_\ell^0), \qquad \boldsymbol{\tau}_\ell^0 = \mathrm{Cov}(Q_\ell^0)$
6: **end for**

7: **for** $k = 0, 1, \ldots$ **do**
8: $\quad$ // Forward Pass
9: $\quad \widehat{Q}_{k0}^+ = h_0^+(Q_{k0}^-, W_0, \overline{\boldsymbol{\Theta}}_{k0}^+)$
10: $\quad \overline{\boldsymbol{\Lambda}}_{k0}^+ = (\mathbb{E}\frac{\partial \widehat{Q}_{k0}^+}{\partial Q_0^-})^{-1}\overline{\boldsymbol{\Gamma}}_{k,0}^-$
11: $\quad \overline{\boldsymbol{\Gamma}}_{k0}^+ = \overline{\boldsymbol{\Lambda}}_{k0}^+ - \overline{\boldsymbol{\Gamma}}_{k0}^-$
12: $\quad Q_{k0}^+ = f_0^+(Q_{k0}^-, W_0, \overline{\boldsymbol{\Omega}}_{k0}^+)$
13: $\quad (P_0^0, P_{k0}^+) \sim \mathcal{N}(\mathbf{0}, \mathbf{K}_{k0}^+), \qquad \mathbf{K}_{k0}^+ := \mathrm{Cov}(Q_0^0, Q_{k0}^+)$

14: $\quad$ **for** $\ell = 1, \ldots, L-1$ **do**
15: $\quad\quad \widehat{Q}_{k\ell}^+ = h_\ell^+(P_{\ell-1}^0, P_{k,\ell-1}^+, Q_{k\ell}^-, W_\ell, \overline{\boldsymbol{\Theta}}_{k\ell}^+)$
16: $\quad\quad \overline{\boldsymbol{\Lambda}}_{k\ell}^+ = (\mathbb{E}\frac{\partial \widehat{Q}_{k\ell}^+}{\partial Q_{k\ell}^-})^{-1}\overline{\boldsymbol{\Gamma}}_{k\ell}^-$
17: $\quad\quad \overline{\boldsymbol{\Gamma}}_{k\ell}^+ = \overline{\boldsymbol{\Lambda}}_{k\ell}^+ - \overline{\boldsymbol{\Gamma}}_{k\ell}^-$
18: $\quad\quad Q_{k\ell}^+ = f_\ell^+(P_{\ell-1}^0, P_{k,\ell-1}^+, Q_{k\ell}^-, W_\ell, \overline{\boldsymbol{\Omega}}_{k\ell}^+)$
19: $\quad\quad (P_\ell^0, P_{k\ell}^+) \sim \mathcal{N}(\mathbf{0}, \mathbf{K}_{k\ell}^+), \quad \mathbf{K}_{k\ell}^+ := \mathrm{Cov}(Q_\ell^0, Q_{k\ell}^+)$
20: $\quad$ **end for**

21: $\quad$ // Backward Pass
22: $\quad \widehat{P}_{k+1,L-1}^- = h_L^-(P_{L-1}^0, P_{k,L-1}^+, W_L, \overline{\boldsymbol{\Theta}}_{k+1,L}^-)$
23: $\quad \overline{\boldsymbol{\Lambda}}_{k+1,L}^- = (\mathbb{E}\frac{\partial \widehat{P}_{k+1,L-1}^-}{\partial P_{L-1}^+})^{-1}\overline{\boldsymbol{\Gamma}}_{kL}^+$
24: $\quad \overline{\boldsymbol{\Gamma}}_{k+1,L-1}^- = \overline{\boldsymbol{\Lambda}}_{k+1,L-1}^- - \overline{\boldsymbol{\Gamma}}_{k,L-1}^+,$
25: $\quad P_{k+1,L-1}^- = f_L^-(P_{L-1}^0, P_{k,L-1}^+, W_L, \overline{\boldsymbol{\Omega}}_{k+1,L}^-)$
26: $\quad Q_{k+1,L-1}^- \sim \mathcal{N}(\mathbf{0}, \boldsymbol{\tau}_{k+1,L-1}^-), \ \boldsymbol{\tau}_{k+1,L-1}^- := \mathrm{Cov}(P_{k+1,L-1}^-)$
27: $\quad$ **for** $\ell = L-2, \ldots, 0$ **do**
28: $\quad\quad \widehat{P}_{k+1,\ell}^- = h_\ell^-(P_\ell^0, P_{k\ell}^+, Q_{k+1,\ell+1}^-, W_\ell, \overline{\boldsymbol{\Theta}}_{k+1,\ell}^-)$
29: $\quad\quad \overline{\boldsymbol{\Lambda}}_{k+1,\ell}^- = (\mathbb{E}\frac{\partial \widehat{P}_{k+1,\ell}^-}{\partial P_{k,\ell}^+})^{-1}\overline{\boldsymbol{\Gamma}}_{k,\ell}^+$
30: $\quad\quad \overline{\boldsymbol{\Gamma}}_{k+1,\ell}^- = \overline{\boldsymbol{\Lambda}}_{k+1,\ell}^- - \overline{\boldsymbol{\Gamma}}_{k,\ell}^+,$
31: $\quad\quad P_{k+1,\ell}^- = f_\ell^-(P_\ell^0, P_{k\ell}^+, Q_{k+1,\ell+1}^-, W_\ell, \overline{\boldsymbol{\Omega}}_{k+1,\ell}^-)$
32: $\quad\quad Q_{k+1,\ell}^- \sim \mathcal{N}(\mathbf{0}, \boldsymbol{\tau}_{k+1,\ell}^-), \quad \boldsymbol{\tau}_{k+1,\ell}^- := \mathrm{Cov}(P_{k+1,\ell}^-)$
33: $\quad$ **end for**
34: **end for**

---

order $p$,

$$\lim_{N\to\infty} \frac{1}{N}\sum_{n=1}^{N} f(\mathbf{X}_{n:}^{(N)}) = \mathbb{E}[f(X)] \quad \text{a.s.} \tag{22}$$

Note that the sequence $\{\mathbf{X}^{(N)}\}$ could be random or deterministic. If it is random, however, then the quantities on the left hand side are random sums and the almost sure convergence must take this randomness into account as well.

The above convergence is equivalent to requiring weak convergence as well as convergence of the $p^{\text{th}}$ moment, of the empirical distribution $\frac{1}{N}\sum_{n=1}^{N}\delta_{\mathbf{X}_{n:}^{(N)}}$ of the rows of $\mathbf{X}^{(N)}$ to $X$. This is also referred to convergence in the Wasserstein-$p$ metric [44, Chap. 6].

In the case of $p = 2$, the condition is equivalent to requiring (22) to hold for all continuously bounded functions $f$ as well as for all $f_q(\boldsymbol{x}) = \boldsymbol{x}^\mathsf{T}\boldsymbol{Q}\boldsymbol{x}$ for all positive definite matrices $\boldsymbol{Q}$.

**Definition 3** (Uniform Lipschitz continuity). For a positive definite matrix $\boldsymbol{M}$, the map $\phi(\mathbf{r};\boldsymbol{M}):$ $\mathbb{R}^d \to \mathbb{R}^d$ is said to be uniformly Lipschitz continuous in $\mathbf{r}$ at $\boldsymbol{M} = \overline{\boldsymbol{M}}$ if there exist non-negative constants $L_1$, $L_2$ and $L_3$ such that for all $\mathbf{r} \in \mathbb{R}^d$

$$\|\phi(\mathbf{r}_1;\boldsymbol{M}_0) - \phi(\mathbf{r}_2;\boldsymbol{M}_0)\| \le L_1\|\mathbf{r}_1 - \mathbf{r}_2\|$$
$$\|\phi(\mathbf{r};\boldsymbol{M}_1) - \phi(\mathbf{r};\boldsymbol{M}_2)\| \le L_2(1 + \|\mathbf{r}\|)\rho(\boldsymbol{M}_1,\boldsymbol{M}_2)$$

for all $\boldsymbol{M}_i$ such that $\rho(\boldsymbol{M}_i,\overline{\boldsymbol{M}}) < L_3$ where $\rho$ is a metric on the cone of positive semidefinite matrices.

We are now ready to prove Theorem 1.

# Appendix C  Proof of Theorem 1

The proof of Theorem 1 is a special case of a more general result on multi-layer recursions given in Theorem 2. This result is stated in Appendix D, and proved in Appendix E. The rest of this section identifies certain relevant quantities from Theorem 1 in order to apply Theorem 2.

Consider the SVD given of weight matrices $\mathbf{W}_\ell$ of the network given by,

$$\mathbf{W}_\ell = \mathbf{V}_\ell \text{diag}(\mathbf{S}_\ell)\mathbf{V}_\ell - 1$$

as explained in Section 4 of the main paper. We analyze Algo. 1 using *transformed* versions of the true signals $\mathbf{Z}_\ell^0$ and input errors $\mathbf{R}_\ell^\pm - \mathbf{Z}_\ell^0$ to the denoisers $\mathbf{G}_\ell^\pm$. For $\ell = 0, 2, \ldots L-2$, define

$$\mathbf{q}_\ell^0 = \mathbf{Z}_\ell^0 \qquad\qquad \mathbf{q}_{\ell+1}^0 = \mathbf{V}_{\ell+1}^\top \mathbf{Z}_{\ell+1}^0 \tag{23a}$$
$$\mathbf{p}_\ell^0 = \mathbf{V}_\ell \mathbf{Z}_\ell^0 \qquad\qquad \mathbf{p}_{\ell+1}^0 = \mathbf{Z}_{\ell+1}^0 \tag{23b}$$

which are depicted in Fig. 3 (TOP). Similarly, define the following *transformed* versions of errors in the inputs $\mathbf{R}_\ell^\pm$ to the denoisers $\mathbf{G}_\ell^\pm$

$$\mathbf{q}_\ell^- = \mathbf{R}_\ell^- - \mathbf{Z}_\ell^0 \qquad\qquad \mathbf{q}_{\ell+1}^- = \mathbf{V}_{\ell+1}^\top(\mathbf{R}_{\ell+1}^- - \mathbf{Z}_{\ell+1}^0) \tag{24a}$$
$$\mathbf{p}_\ell^+ = \mathbf{V}_\ell(\mathbf{R}_\ell^+ - \mathbf{Z}_\ell^0) \qquad\qquad \mathbf{p}_{\ell+1}^+ = \mathbf{R}_{\ell+1}^+ - \mathbf{Z}_{\ell+1}^0 \tag{24b}$$

These quantities are depicted as inputs to function blocks $\mathbf{f}_\ell^\pm$ in Fig. 3 (MIDDLE). Define perturbation variables $\mathbf{w}_\ell$ as

$$\mathbf{w}_0 = \mathbf{Z}_0^0, \;\; \mathbf{w}_L = (\mathbf{Y}, \boldsymbol{\Xi}_L), \;\; \mathbf{w}_\ell = \boldsymbol{\Xi}_\ell, \qquad\qquad \forall \ell \in \mathcal{L}_{\text{even}} \tag{25a}$$
$$\mathbf{w}_\ell = (\mathbf{S}_\ell, \overline{\mathbf{B}}_\ell, \boldsymbol{\Xi}_\ell), \qquad\qquad \forall \ell \in \mathcal{L}_{\text{odd}} \tag{25b}$$

Finally, we define $\mathbf{q}_\ell^+$ and $\mathbf{p}_\ell^-$ for $\ell = 1, 2, \ldots, L-1$ as

$$\mathbf{q}_\ell^+ = \mathbf{f}_\ell^+(\mathbf{p}_{\ell-1}^0, \mathbf{p}_{\ell-1}^+, \mathbf{q}_\ell^-, \mathbf{w}_\ell, \Omega_\ell) \tag{26a}$$
$$\mathbf{p}_{\ell-1}^- = \mathbf{f}_\ell^-(\mathbf{p}_{\ell-1}^0, \mathbf{p}_{\ell-1}^+, \mathbf{q}_\ell^-, \mathbf{w}_\ell, \Omega_\ell), \tag{26b}$$

which are outputs of function blocks in Fig. 3 (MIDDLE). Similarly, define the quantities $\mathbf{q}_0^+ = \mathbf{f}_0^+(\mathbf{q}_0^-, \mathbf{Z}_0, \Omega_0)$ and $\mathbf{p}_{L-1}^- = \mathbf{f}_L^+(\mathbf{p}_{L-1}^0, \mathbf{p}_{L-1}^+, \mathbf{Y}, \Omega_L)$.

**Lemma 1.** *Algorithm 1 is a special case of Algorithm 3 with the definitions $\{\mathbf{q}_\ell^0, \mathbf{p}_\ell^0, \mathbf{q}_\ell^\pm, \mathbf{p}_\ell^\pm\}_{\ell=0}^{L-1}$ given in equations (23),(24), and (26), functions $\mathbf{f}_\ell^\pm$ are row-wise extensions of $f_\ell^\pm$ defined using equations (21) and (20).*

**Lemma 2.** *Assumptions 1 and 2 required for applying Theorem 2 are satisfied by the conditions in Theorem 1.*

*Proof.* The proofs of the above lemmas are identical to the case of $d = 1$, which was shown in [34]. For details see [34, Appendix F]. □

## Appendix D    General Multi-Layer Recursions

Figure 3:  (TOP) The equations (1) with equivalent quantities defined in (23), and $\mathbf{f}_\ell^0$ defined using (19).
(MIDDLE) The Gen-ML-Mat recursions in Algorithm 3. These are also equivalent to ML-Mat-VAMP recursions from Algorithm 1 (See Lemma 1) if $\mathbf{q}^\pm, \mathbf{p}^\pm$ are as defined as in equations (24) and (26), and $\mathbf{f}_\ell^\pm$ given by equations (21) and (20).
(BOTTOM) Quantities in the GEN-ML-SE recursions. These are also equivalent to ML-Mat-VAMP SE recursions from Algorithm 2 (See Lemma 1)
The iteration indices $k$ have been dropped for notational simplicity.

To analyze Algorithm 1, we consider a more general class of recursions as given in Algorithm 3 and depicted in Fig. 3. The Gen-ML recursions generates (i) a set of *true matrices* $\mathbf{q}_\ell^0$ and $\mathbf{p}_\ell^0$ and (ii) *iterated matrices* $\mathbf{q}_{k\ell}^\pm$ and $\mathbf{p}_{k\ell}^\pm$. Each of these matrices have the same number of columns, denoted by $d$.

The true matrices are generated by a single forward pass, whereas the iterated matrices are generated via a sequence of forward and backward passes through a multi-layer system. In proving the State Evolution for the ML-Mat-VAMP algorithm (Algo. 1, one would then associate the terms $\mathbf{q}_{k\ell}^\pm$ and $\mathbf{p}_{k\ell}^\pm$ with certain error quantities in the ML-Mat-VAMP recursions. To account for the effect of the parameters $\mathbf{\Gamma}_{k\ell}^\pm$ and $\mathbf{\Lambda}_{k\ell}^\pm$ in ML-Mat-VAMP, the Gen-ML algorithm describes the parameter updates through a sequence of *parameter lists* $\Upsilon_{k\ell}^\pm$. The parameter lists are ordered lists of parameters that accumulate as the algorithm progresses. The true and iterated matrices from Algorithm 3 are depicted in the signal flow graphs on the (TOP) and (MIDDLE) panel of Fig. 3 respectively. The iteration index $k$ for the iterated vectors $\mathbf{q}_{k\ell}, \mathbf{p}_{k\ell}$ has been dropped for simplifying notation.

The functions $\mathbf{f}_\ell^0(\cdot)$ that produce the true matrices $\mathbf{q}_\ell^0, \mathbf{p}_\ell^0$ are called *initial matrix functions* and use the initial parameter list $\Upsilon_{01}^-$. The functions $\mathbf{f}_{k\ell}^\pm(\cdot)$ that produce the matrices $\mathbf{q}_{k\ell}^+$ and $\mathbf{p}_{k\ell}^-$ are called the *matrix update functions* and use parameter lists $\Upsilon_{kl}^\pm$. The initial parameter lists $\Upsilon_{01}^-$ are assumed to be provided. As the algorithm progresses, new parameters $\lambda_{k\ell}^\pm$ are computed and then added to the lists in lines 12, 18, 25 and 31. The matrix update functions $\mathbf{f}_{k\ell}^\pm(\cdot)$ may depend on any sets of parameters accumulated in the parameter list. In lines 11, 17, 24 and 30, the new parameters $\lambda_{k\ell}^\pm$ are computed by: (1) computing average values $\mu_{k\ell}^\pm$ of *row-wise* functions $\boldsymbol{\varphi}_{k\ell}^\pm(\cdot)$; and (2) taking functions $T_{k\ell}^\pm(\cdot)$ of the average values $\mu_{k\ell}^\pm$. Since the average values $\mu_{k\ell}^\pm$ represent statistics on the rows of $\boldsymbol{\varphi}_{k\ell}^\pm(\cdot)$, we will call $\boldsymbol{\varphi}_{k\ell}^\pm(\cdot)$ the *parameter statistic functions*. We will call the $T_{k\ell}^\pm(\cdot)$ the *parameter update functions*. The functions $\mathbf{f}_\ell^0, \mathbf{f}_{k\ell}^\pm, \boldsymbol{\varphi}_\ell^\pm$ also take as input some perturbation vectors $\mathbf{w}_\ell$.

Similar to the analysis of the ML-Mat-VAMP Algorithm, we consider the following large-system limit (LSL) analysis of Gen-ML. Specifically, we consider a sequence of runs of the recursions indexed by $N$. For each $N$, let $N_\ell = N_\ell(N)$ be the dimension of the matrix signals $\mathbf{p}_\ell^\pm$ and $\mathbf{q}_\ell^\pm$ as we assume that $\lim_{N \to \infty} \frac{N_\ell}{N} = \beta_\ell \in (0, \infty)$ is a constant so that $N_\ell$ scales linearly with $N$. Note however that the number of columns of each of the matrices $\{\mathbf{q}_\ell^0, \mathbf{p}_\ell^0, \mathbf{q}_{k\ell}^\pm, \mathbf{p}_{k\ell}^\pm\}$ is equal to a finite integer $d > 0$, which remains fixed for all $N$. We then make the following assumptions. See Appendix B for an overview of empirical convergence of sequences which we use in the assumptions described below.

**Assumption 1.** For vectors in the Gen-ML Algorithm (Algorithm 3), we assume:

(a) The matrices $\mathbf{V}_\ell$ are Haar distributed on the set of $N_\ell \times N_\ell$ orthogonal matrices and are independent from one another and from the matrices $\mathbf{q}_0^0$, $\mathbf{q}_{0\ell}^-$, perturbation variables $\mathbf{w}_\ell$.

(b) The rows of the initial matrices $\mathbf{q}_{0\ell}^-$, and perturbation variables $\mathbf{w}_\ell$ converge jointly empirically with limits,

$$\mathbf{q}_{0\ell}^- \overset{2}{\Rightarrow} Q_{0\ell}^-, \quad \mathbf{w}_\ell \overset{2}{\Rightarrow} W_\ell, \tag{27}$$

where $Q_{0\ell}^-$ are random vectors in $\mathbb{R}^{1 \times d}$ such that $(Q_{00}^-, \cdots, Q_{0,L-1}^-)$ is jointly Gaussian. For $\ell = 0, \ldots, L-1$, the random variables $W_\ell, P_{\ell-1}^0$ and $Q_{0\ell}^-$ are all independent. We also assume that the initial parameter list converges as

$$\lim_{N \to \infty} \Upsilon_{01}^-(N) \xrightarrow{a.s.} \overline{\Upsilon}_{01}^-, \tag{28}$$

to some list $\overline{\Upsilon}_{01}^-$. The limit (28) means that every element in the list $\lambda(N) \in \Upsilon_{01}^-(N)$ converges to a limit $\lambda(N) \to \overline{\lambda} \in \overline{\Upsilon}_{01}^-$ as $N \to \infty$ almost surely.

(c) The *matrix update functions* $\mathbf{f}_{k\ell}^\pm(\cdot)$ and *parameter update functions* $\boldsymbol{\varphi}_{k\ell}^\pm(\cdot)$ act row-wise. For e.g., in the $k^{\text{th}}$ forward pass, at stage $\ell$, we assume that for each output row $n$,

$$\left[\mathbf{f}_{k\ell}^+(\mathbf{p}_{\ell-1}^0, \mathbf{p}_{k,\ell-1}^+, \mathbf{q}_{k\ell}^-, \mathbf{w}_\ell, \Upsilon_{k\ell}^+)\right]_{n:} = f_{k\ell}^+(\mathbf{p}_{\ell-1,n:}^0, \mathbf{p}_{k,\ell-1,n:}^+, \mathbf{q}_{k\ell,n:}^-, \mathbf{w}_{\ell,n:}, \Upsilon_{k\ell}^+)$$

$$\left[\boldsymbol{\varphi}_{k\ell}^+(\mathbf{p}_{\ell-1}^0, \mathbf{p}_{k,\ell-1}^+, \mathbf{q}_{k\ell}^-, \mathbf{w}_\ell, \Upsilon_{k\ell}^+)\right]_{n:} = \varphi_{k\ell}^+(\mathbf{p}_{\ell-1,n:}^0, \mathbf{p}_{k,\ell-1,n:}^+, \mathbf{q}_{k\ell,n:}^-, \mathbf{w}_{\ell,n:}, \Upsilon_{k\ell}^+),$$

for some $\mathbb{R}^{1 \times d}$-valued functions $f_{k\ell}^+(\cdot)$ and $\varphi_{k\ell}^+(\cdot)$. Similar definitions apply in the reverse directions and for the initial vector functions $\mathbf{f}_\ell^0(\cdot)$. We will call $f_{k\ell}^\pm(\cdot)$ the *matrix update row-wise functions* and $\varphi_{k\ell}^\pm(\cdot)$ the *parameter update row-wise functions*.

Next we define a set of *deterministic* constants $\{\mathbf{K}_{k\ell}^+, \boldsymbol{\tau}_{k\ell}^-, \overline{\mu}_{k\ell}^\pm, \overline{\Upsilon}_{kl}^\pm, \boldsymbol{\tau}_\ell^0\}$ and $\mathbb{R}^{1 \times d}$-valued random vectors $\{Q_\ell^0, P_\ell^0, Q_{k\ell}^\pm, P_\ell^\pm\}$ which are recursively defined through Algorithm 4, which we call the *Gen-ML-Mat State Evolution* (SE). These recursions in Algorithm closely mirror those in the Gen-ML-Mat algorithm (Algorithm 3). The matrices $\mathbf{q}_{k\ell}^\pm$ and $\mathbf{p}_{k\ell}^\pm$ are replaced by random vectors $Q_{k\ell}^\pm$ and $P_{k\ell}^\pm$; the matrix and parameter update functions $\mathbf{f}_{k\ell}^\pm(\cdot)$ and $\boldsymbol{\varphi}_{k\ell}^\pm(\cdot)$ are replaced by their row-wise functions $f_{k\ell}^\pm(\cdot)$ and $\varphi_{k\ell}^\pm(\cdot)$; and the parameters $\lambda_{k\ell}^\pm$ are replaced by their limits $\overline{\lambda}_{k\ell}^\pm$. We refer to $\{Q_\ell^0, P_\ell^0\}$ as *true random vectors* and $\{Q_{k\ell}^\pm, P_{kl}^\pm\}$ as *iterated random vectors*. The signal flow graph for the true and iterated random variables in Algorithm 4 is given in the (BOTTOM) panel of Fig. 3. The iteration index $k$ for the iterated random variables $\{Q_{k\ell}^\pm, P_{kl}^\pm\}$ to simplify notation.

---

**Algorithm 3** General Multi-Layer Matrix (Gen-ML-Mat) Recursion

---

**Require:** Initial matrix functions $\{\mathbf{f}_\ell^0\}$. Matrix update functions $\{\mathbf{f}_{k\ell}^\pm(\cdot)\}$. Parameter statistic functions $\{\boldsymbol{\varphi}_{k\ell}^\pm(\cdot)\}$. Parameter update functions $\{T_{k\ell}^\pm(\cdot)\}$. Orthogonal matrices $\{\mathbf{V}_\ell\}$. Perturbation variables $\{\mathbf{w}_\ell^\pm\}$. Initial matrices $\{\mathbf{q}_{0\ell}^-\}$. Initial parameter list $\Upsilon_{01}^-$.

1: // Initial Pass
2: $\mathbf{q}_0^0 = \mathbf{f}_0^0(\mathbf{w}_0), \quad \mathbf{p}_0^0 = \mathbf{V}_0\mathbf{q}_0^0$
3: **for** $\ell = 1, \dots, L-1$ **do**
4: $\quad \mathbf{q}_\ell^0 = \mathbf{f}_\ell^0(\mathbf{p}_{\ell-1}^0, \mathbf{w}_\ell, \Upsilon_{01}^-)$
5: $\quad \mathbf{p}_\ell^0 = \mathbf{V}_\ell\mathbf{q}_\ell^0$
6: **end for**
7:
8: **for** $k = 0, 1, \dots$ **do**
9: $\quad$ // Forward Pass
10: $\quad \lambda_{k0}^+ = T_{k0}^+(\mu_{k0}^+, \Upsilon_{0k}^-)$
11: $\quad \mu_{k0}^+ = \left\langle \boldsymbol{\varphi}_{k0}^+(\mathbf{q}_{k0}^-, \mathbf{w}_0, \Upsilon_{0k}^-) \right\rangle$
12: $\quad \Upsilon_{k0}^+ = (\Upsilon_{k1}^-, \lambda_{k0}^+)$
13: $\quad \mathbf{q}_{k0}^+ = \mathbf{f}_{k0}^+(\mathbf{q}_{k0}^-, \mathbf{w}_0, \Upsilon_{k0}^+)$
14: $\quad \mathbf{p}_{k0}^+ = \mathbf{V}_0\mathbf{q}_{k0}^+$
15: $\quad$ **for** $\ell = 1, \dots, L-1$ **do**
16: $\quad\quad \lambda_{k\ell}^+ = T_{k\ell}^+(\mu_{k\ell}^+, \Upsilon_{k,\ell-1}^+)$
17: $\quad\quad \mu_{k\ell}^+ = \left\langle \boldsymbol{\varphi}_{k\ell}^+(\mathbf{p}_{\ell-1}^0, \mathbf{p}_{k,\ell-1}^+, \mathbf{q}_{k\ell}^-, \mathbf{w}_\ell, \Upsilon_{k,\ell-1}^+) \right\rangle$
18: $\quad\quad \Upsilon_{k\ell}^+ = (\Upsilon_{k,\ell-1}^+, \lambda_{k\ell}^+)$
19: $\quad\quad \mathbf{q}_{k\ell}^+ = \mathbf{f}_{k\ell}^+(\mathbf{p}_{\ell-1}^0, \mathbf{p}_{k,\ell-1}^+, \mathbf{q}_{k\ell}^-, \mathbf{w}_\ell, \Upsilon_{k\ell}^+)$
20: $\quad\quad \mathbf{p}_{k\ell}^+ = \mathbf{V}_\ell\mathbf{q}_{k\ell}^+$
21: $\quad$ **end for**

22: $\quad$ // Backward Pass
23: $\quad \lambda_{k+1,L}^- = T_{kL}^-(\mu_{kL}^-, \Upsilon_{k,L-1}^+)$
24: $\quad \mu_{kL}^- = \left\langle \boldsymbol{\varphi}_{kL}^-(\mathbf{p}_{k,L-1}^+, \mathbf{w}_L, \Upsilon_{k,L-1}^+) \right\rangle$
25: $\quad \Upsilon_{k+1,L}^- = (\Upsilon_{k,L-1}^+, \lambda_{k+1,L}^+)$
26: $\quad \mathbf{p}_{k+1,L-1}^- = \mathbf{f}_{kL}^-(\mathbf{p}_{L-1}^0, \mathbf{p}_{k,L-1}^+, \mathbf{w}_L, \Upsilon_{k+1,L}^-)$
27: $\quad \mathbf{q}_{k+1,L-1}^- = \mathbf{V}_{L-1}^\mathsf{T}\mathbf{p}_{k+1,L-1}$
28: $\quad$ **for** $\ell = L-1, \dots, 1$ **do**
29: $\quad\quad \lambda_{k+1,\ell}^- = T_{k\ell}^-(\mu_{k\ell}^-, \Upsilon_{k+1,\ell+1}^-)$
30: $\quad\quad \mu_{k\ell}^- = \left\langle \boldsymbol{\varphi}_{k\ell}^-(\mathbf{p}_{\ell-1}^0, \mathbf{p}_{k,\ell-1}^+, \mathbf{q}_{k+1,\ell}^-, \mathbf{w}_\ell, \Upsilon_{k+1,\ell+1}^-) \right\rangle$
31: $\quad\quad \Upsilon_{k+1,\ell}^- = (\Upsilon_{k+1,\ell+1}^-, \lambda_{k+1,\ell}^-)$
32: $\quad\quad \mathbf{p}_{k+1,\ell-1}^- = \mathbf{f}_{k\ell}^-(\mathbf{p}_{\ell-1}^0, \mathbf{p}_{k,\ell-1}^+, \mathbf{q}_{k+1,\ell}^-, \mathbf{w}_\ell, \Upsilon_{k+1,\ell}^-)$
33: $\quad\quad \mathbf{q}_{k+1,\ell-1}^- = \mathbf{V}_{\ell-1}^\mathsf{T}\mathbf{p}_{k+1,\ell-1}^-$
34: $\quad$ **end for**
35: **end for**

---

We also assume the following about the behaviour of row-wise functions around the quantities defined in Algorithm 4. The iteration index $k$ has been dropped for simplifying notation.

**Assumption 2.** For row-wise functions $f, \varphi$ and parameter update functions $T$ we assume:

(a) $T_{k\ell}^\pm(\mu_{k\ell}^\pm, \cdot)$ are continuous at $\mu_{k\ell}^\pm = \overline{\mu}_{k\ell}^\pm$

(b) $f_{k\ell}^+(p_{\ell-1}^0, p_{k,\ell-1}^+, q_{k\ell}^-, w_\ell, \Upsilon_{k\ell}^+)$, $\quad\quad \frac{\partial f_{k\ell}^+}{\partial q_{k\ell}^-}(p_{\ell-1}^0, p_{k,\ell-1}^+, q_{k\ell}^-, w_\ell, \Upsilon_{k\ell}^+) \quad$ and
$\varphi_{k\ell}^+(p_{\ell-1}^0, p_{k,\ell-1}^+, q_{k\ell}^-, w_\ell, \Upsilon_{k,\ell-1}^+)$ are uniformly Lipschitz continuous in $(p_{\ell-1}^0, p_{k,\ell-1}^+, q_{k\ell}^-, w_\ell)$

---

**Algorithm 4** Gen-ML-Mat State Evolution (SE)

---

**Require:** Matrix update row-wise functions $f_\ell^0(\cdot)$ and $f_{k\ell}^\pm(\cdot)$, parameter statistic row-wise functions $\varphi_{k\ell}^\pm(\cdot)$, parameter update functions $T_{k\ell}^\pm(\cdot)$, initial parameter list limit: $\overline{\Upsilon}_{01}^-$, initial random variables $W_\ell, Q_{0\ell}^-, \ell = 0, \ldots, L-1$.

1: // Initial pass
2: $Q_0^0 = f_0^0(W_0, \overline{\Upsilon}_{01}^-), \quad P_0^0 \sim \mathcal{N}(0, \tau_0^0), \quad \tau_0^0 = \mathbb{E}(Q_0^0)^2$
3: **for** $\ell = 1, \ldots, L-1$ **do**
4: $\quad Q_\ell^0 = f_\ell^0(P_{\ell-1}^0, W_\ell, \overline{\Upsilon}_{01}^-)$
5: $\quad P_\ell^0 \sim \mathcal{N}(0, \tau_\ell^0), \quad \tau_\ell^0 = \mathrm{Cov}(Q_\ell^0)$
6: **end for**
7:
8: **for** $k = 0, 1, \ldots$ **do**
9: $\quad$ // Forward Pass
10: $\quad \overline{\lambda}_{k0}^+ = T_{k0}^+(\overline{\mu}_{k0}^+, \overline{\Upsilon}_{0k}^-)$
11: $\quad \overline{\mu}_{k0}^+ = \mathbb{E}(\varphi_{k0}^+(Q_{k0}^-, W_0, \overline{\Upsilon}_{0k}^-))$
12: $\quad \overline{\Upsilon}_{k0}^+ = (\overline{\Upsilon}_{k1}^-, \overline{\lambda}_{k0}^+)$
13: $\quad Q_{k0}^+ = f_{k0}^+(Q_{k0}^-, W_0, \overline{\Upsilon}_{k0}^+)$
14: $\quad (P_0^0, P_{k0}^+) \sim \mathcal{N}(\mathbf{0}, \mathbf{K}_{k0}^+), \quad \mathbf{K}_{k0}^+ = \mathrm{Cov}(Q_0^0, Q_{k0}^+)$
15: $\quad$ **for** $\ell = 1, \ldots, L-1$ **do**
16: $\quad\quad \overline{\lambda}_{k\ell}^+ = T_{k\ell}^+(\overline{\mu}_{k\ell}^+, \overline{\Upsilon}_{k,\ell-1}^+)$
17: $\quad\quad \overline{\mu}_{k\ell}^+ = \mathbb{E}(\varphi_{k\ell}^+(P_{\ell-1}^0, P_{k,\ell-1}^+, Q_{k\ell}^-, W_\ell, \overline{\Upsilon}_{k,\ell-1}^+))$
18: $\quad\quad \overline{\Upsilon}_{k\ell}^+ = (\overline{\Upsilon}_{k,\ell-1}^+, \overline{\lambda}_{k\ell}^+)$
19: $\quad\quad Q_{k\ell}^+ = f_{k\ell}^+(P_{\ell-1}^0, P_{k,\ell-1}^+, Q_{k\ell}^-, W_\ell, \overline{\Upsilon}_{k\ell}^+)$
20: $\quad\quad (P_\ell^0, P_{k\ell}^+) \sim \mathcal{N}(\mathbf{0}, \mathbf{K}_{k\ell}^+), \quad \mathbf{K}_{k\ell}^+ = \mathrm{Cov}(Q_\ell^0, Q_{k\ell}^+)$
21: $\quad$ **end for**
22: $\quad$ // Backward Pass
23: $\quad \overline{\lambda}_{k+1,L}^- = T_{kL}^-(\overline{\mu}_{kL}^-, \overline{\Upsilon}_{k,L-1}^+)$
24: $\quad \overline{\mu}_{kL}^- = \mathbb{E}(\varphi_{kL}^-(P_{L-1}^0, P_{k,L-1}^+, W_L, \overline{\Upsilon}_{k,L-1}^+))$
25: $\quad \overline{\Upsilon}_{k+1,L}^- = (\overline{\Upsilon}_{k,L-1}^+, \overline{\lambda}_{k+1,L}^-)$
26: $\quad P_{k+1,L-1}^- = f_{kL}^-(P_{L-1}^0, P_{k,L-1}^+, W_L, \overline{\Upsilon}_{k+1,L}^-)$
27: $\quad Q_{k+1,L-1}^- \sim \mathcal{N}(0, \tau_{k+1,L-1}^-), \quad \tau_{k+1,L-1}^- = \mathrm{Cov}(P_{k+1,L-1}^-)$
28: $\quad$ **for** $\ell = L-1, \ldots, 1$ **do**
29: $\quad\quad \overline{\lambda}_{k+1,\ell}^- = T_{k\ell}^-(\overline{\mu}_{k\ell}^-, \overline{\Upsilon}_{k+1,\ell+1}^-)$
30: $\quad\quad \overline{\mu}_{k\ell}^- = \mathbb{E}(\varphi_{k\ell}^-(P_{\ell-1}^0, P_{k,\ell-1}^+, Q_{k+1,\ell}^-, W_\ell, \overline{\Upsilon}_{k+1,\ell+1}^-))$
31: $\quad\quad \overline{\Upsilon}_{k+1,\ell}^- = (\overline{\Upsilon}_{k+1,\ell+1}^-, \overline{\lambda}_{k+1,\ell}^-)$
32: $\quad\quad P_{k+1,\ell-1}^- = f_{k\ell}^-(P_{\ell-1}^0, P_{k,\ell-1}^+, Q_{k+1,\ell}^-, W_\ell, \overline{\Upsilon}_{k+1,\ell}^-)$
33: $\quad\quad Q_{k+1,\ell-1}^- \sim \mathcal{N}(0, \tau_{k+1,\ell-1}^-), \quad \tau_{k+1,\ell-1}^- = \mathrm{Cov}(P_{k+1,\ell-1}^-)$
34: $\quad$ **end for**
35: **end for**

---

at $\Upsilon_{k\ell}^+ = \overline{\Upsilon}_{k\ell}^+, \Upsilon_{k,\ell-1}^+ = \overline{\Upsilon}_{k,\ell-1}^+$. Similarly,

$$f_{k+1,\ell}^-(p_{\ell-1}^0, p_{k,\ell-1}^+, q_{k+1,\ell}^-, w_\ell, \Upsilon_{k\ell}^-), \qquad \frac{\partial f_{k\ell}^-}{\partial p_{k,\ell-1}^+}(p_{\ell-1}^0, p_{k,\ell-1}^+, q_{k+1,\ell}^-, w_\ell, \Upsilon_{k\ell}^-), \qquad \text{and}$$

$\varphi_{k\ell}^-(p_{\ell-1}^0, p_{k,\ell-1}^+, q_{k+1,\ell}^-, w_\ell, \Upsilon_{k+1,\ell+1}^-)$ are uniformly Lipschitz continuous in $(p_{\ell-1}^0, p_{k,\ell-1}^+, q_{k+1,\ell}^-, w_\ell)$ at $\Upsilon_{k\ell}^- = \overline{\Upsilon}_{k\ell}^-, \Upsilon_{k+1,\ell+1}^- = \overline{\Upsilon}_{k+1,\ell+1}^-$.

(c) $f_\ell^0(p_{\ell-1}^0, w_\ell, \Upsilon_{01}^-)$ are uniformly Lipschitz continuous in $(p_{k,\ell-1}^0, w_\ell)$ at $\Upsilon_{k+1,\ell}^- = \overline{\Upsilon}_{k+1,\ell}^-$.

(d) Matrix update functions $\mathbf{f}_{k\ell}^{\pm}$ are *asymptotically divergence free* meaning

$$\lim_{N\to\infty}\left\langle\frac{\partial\mathbf{f}_{k\ell}^+}{\partial\mathbf{q}_{k\ell}^-}(\mathbf{p}_{k,\ell-1}^+,\mathbf{q}_{k\ell}^-,\mathbf{w}_\ell,\overline{\Upsilon}_{k\ell}^+)\right\rangle=\mathbf{0},\quad\lim_{N\to\infty}\left\langle\frac{\partial\mathbf{f}_{k\ell}^-}{\partial\mathbf{p}_{k,\ell-1}^+}(\mathbf{p}_{k,\ell-1}^+,\mathbf{q}_{k+1,\ell}^-,\mathbf{w}_\ell,\overline{\Upsilon}_{k\ell}^-)\right\rangle=\mathbf{0}$$

(29)

We are now ready to state the general result regarding the empirical convergence of the true and iterated vectors from Algorithm 3 in terms of random variables defined in Algorithm 4.

**Theorem 2.** *Consider the iterates of the Gen-ML recursion (Algorithm 3) and the corresponding random variables and parameter limits defined by the SE recursions (Algorithm 4) under Assumptions 1 and 2. Then,*

*(a) For any fixed $k \geq 0$ and fixed $\ell = 1, \ldots, L-1$, the parameter list $\Upsilon_{k\ell}^+$ converges as*

$$\lim_{N\to\infty}\Upsilon_{k\ell}^+=\overline{\Upsilon}_{k\ell}^+$$

(30)

*almost surely. Also, the rows of $\mathbf{w}_\ell, \mathbf{p}_{\ell-1}^0, \mathbf{q}_\ell^0, \mathbf{p}_{0,\ell-1}^+, \ldots, \mathbf{p}_{k,\ell-1}^+$ and $\mathbf{q}_{0\ell}^{\pm}, \ldots, \mathbf{q}_{k\ell}^{\pm}$ almost surely jointly converge empirically with limits,*

$$(\mathbf{p}_{\ell-1}^0,\mathbf{p}_{i,\ell-1}^+,\mathbf{q}_{j\ell}^-,\mathbf{q}_\ell^0,\mathbf{q}_{j\ell}^+)\overset{2}{\Rightarrow}(P_{\ell-1}^0,P_{i,\ell-1}^+,Q_{j\ell}^-,Q_\ell^0,Q_{j\ell}^+),$$

(31)

*for all $0 \leq i, j \leq k$, where the variables $P_{\ell-1}^0$, $P_{i,\ell-1}^+$ and $Q_{j\ell}^-$ are zero-mean jointly Gaussian random variables independent of $W_\ell$ and with covariance matrix given by*

$$\mathrm{Cov}(P_{\ell-1}^0,P_{i,\ell-1}^+)=\mathbf{K}_{i,\ell-1}^+,\quad\mathbb{E}(Q_{j\ell}^-)^2=\boldsymbol{\tau}_{j\ell}^-,\quad\mathbb{E}(P_{i,\ell-1}^{+\mathsf{T}}Q_{j\ell}^-)=\mathbf{0},\quad\mathbb{E}(P_{\ell-1}^{0\mathsf{T}}Q_{j\ell}^-)=\mathbf{0},$$

(32)

*and $Q_\ell^0$, $Q_{j\ell}^+$ are the random variable in lines 4, 19, i.e.,*

$$Q_\ell^0=f_\ell^0(P_{\ell-1}^0,W_\ell),\quad Q_{j\ell}^+=f_{j\ell}^+(P_{\ell-1}^0,P_{j,\ell-1}^+,Q_{j\ell}^-,W_\ell,\overline{\Upsilon}_{j\ell}^+).$$

(33)

*An identical result holds for $\ell = 0$ with all the variables $\mathbf{p}_{i,\ell-1}^+$ and $P_{i,\ell-1}^+$ removed.*

*(b) For any fixed $k \geq 1$ and fixed $\ell = 1, \ldots, L-1$, the parameter lists $\Upsilon_{k\ell}^-$ converge as*

$$\lim_{N\to\infty}\Upsilon_{k\ell}^-=\overline{\Upsilon}_{k\ell}^-$$

(34)

*almost surely. Also, the rows of $\mathbf{w}_\ell, \mathbf{p}_{\ell-1}^0, \mathbf{p}_{0,\ell-1}^{\pm}, \ldots, \mathbf{p}_{k-1,\ell-1}^{\pm}$, and $\mathbf{q}_{0\ell}^-, \ldots, \mathbf{q}_{k\ell}^-$ almost surely jointly converge empirically with limits,*

$$(\mathbf{p}_{\ell-1}^0,\mathbf{p}_{i,\ell-1}^+,\mathbf{q}_{j\ell}^-,\mathbf{p}_{j,\ell-1}^-)\overset{2}{\Rightarrow}(P_{\ell-1}^0,P_{i,\ell-1}^+,Q_{j\ell}^-,P_{j,\ell-1}^-),$$

(35)

*for all $0 \leq i \leq k-1$ and $0 \leq j \leq k$, where the variables $P_{\ell-1}^0$, $P_{i,\ell-1}^+$ and $Q_{j\ell}^-$ are zero-mean jointly Gaussian random variables independent of $W_\ell$ and with covariance matrix given by equation (32) and $P_{j\ell}^-$ is the random variable in line 32:*

$$P_{j\ell}^-=f_{j\ell}^-(P_{\ell-1}^0,P_{j-1,\ell-1}^+,Q_{j\ell}^-,W_\ell,\overline{\Upsilon}_{j\ell}^-).$$

(36)

*An identical result holds for $\ell = L$ with all the variables $\mathbf{q}_{j\ell}^-$ and $Q_{j\ell}^-$ removed.*

*For $k = 0$, $\Upsilon_{01}^- \to \overline{\Upsilon}_{01}^-$ almost surely, and the rows $\{(\mathbf{w}_{\ell,n:},\mathbf{p}_{\ell-1,n:}^0,\mathbf{q}_{j\ell,n:}^-)\}_{n=1}^N$ empirically converge to independent random variables $(W_\ell,P_{\ell-1}^0,Q_{0\ell}^-)$.*

*Proof.* Appendix E is dedicated to proving this result. □

# Appendix E    Proof of Theorem 2

The proof proceeds using mathematical induction. It largely mimics the proof for the case of $d = 1$ which were the convergence results in [34, Thm. 5]. However, in the case of $d > 1$, we observe that several quantities which were scalars in proving [34, Thm. 5] are now matrices. Due to the non-commutativity of these matrix quantities, we re-state the whole prove, while modifying the requisite matrix terms appropriately.

### Appendix E.1 Overview of the Induction Sequence

The proof is similar to that of [36, Theorem 4], which provides a SE analysis for VAMP on a single-layer network. The critical challenge here is to extend that proof to multi-layer recursions. Many of the ideas in the two proofs are similar, so we highlight only the key differences between the two.

Similar to the SE analysis of VAMP in [36], we use an induction argument. However, for the multi-layer proof, we must index over both the iteration index $k$ and layer index $\ell$. To this end, let $\mathcal{H}_{k\ell}^+$ and $\mathcal{H}_{k\ell}^-$ be the hypotheses:

- $\mathcal{H}_{k\ell}^+$: The hypothesis that Theorem 2(a) is true for a given $k$ and $\ell$, where $0 \le \ell \le L - 1$.
- $\mathcal{H}_{k\ell}^-$: The hypothesis that Theorem 2(b) is true for a given $k$ and $\ell$, where $1 \le \ell \le L$.

We prove these hypotheses by induction via a sequence of implications,

$$\{\mathcal{H}_{0\ell}^-\}_{\ell=1}^L \cdots \Rightarrow \mathcal{H}_{k1}^- \Rightarrow \mathcal{H}_{k0}^+ \Rightarrow \cdots \Rightarrow \mathcal{H}_{k,L-1}^+ \Rightarrow \mathcal{H}_{k+1,L}^- \Rightarrow \cdots \Rightarrow \mathcal{H}_{k+1,1}^- \Rightarrow \cdots , \qquad (37)$$

beginning with the hypotheses $\{\mathcal{H}_{0\ell}^-\}$ for all $\ell = 1, \ldots, L-1$.

### Appendix E.2 Base Case: Proof of $\{\mathcal{H}_{0\ell}^-\}_{\ell=1}^L$

The base case corresponds to the hypotheses $\{\mathcal{H}_{0\ell}^-\}_{\ell=1}^L$. Note that Theorem 2(b) states that for $k = 0$, we need $\Upsilon_{01}^- \to \overline{\Upsilon}_{01}^-$ almost surely, and $\{(\mathbf{w}_{\ell,n:}, \mathbf{p}_{\ell-1,n:}^0, \mathbf{q}_{j\ell,n:}^-)\}_{n=1}^N$ empirically converge to independent random variables $(W_\ell, P_{\ell-1}^0, Q_{0\ell}^-)$. These follow directly from equations (27) and (28) in Assumption 1 (a).

### Appendix E.3 Inductive Step: Proof of $\mathcal{H}_{k,\ell+1}^+$

Fix a layer index $\ell = 1, \ldots, L-1$ and an iteration index $k = 0, 1, \ldots$. We show the implication $\cdots \implies \mathcal{H}_{k,\ell+1}^+$ in (37). All other implications can be proven similarly using symmetry arguments.

**Definition 4** (Induction hypothesis). The hypotheses prior to $\mathcal{H}_{k,\ell+1}^+$ in the sequence (37), but not including $\mathcal{H}_{k,\ell+1}^+$, are true.

The inductive step then corresponds to the following result.

**Lemma 3.** *Under the induction hypothesis, $\mathcal{H}_{k,\ell+1}^+$ holds*

Before proving the inductive step in Lemma 3, we prove two intermediate lemmas. Let us start by defining some notation. Define $\mathbf{P}_{k\ell}^+ := \begin{bmatrix} \mathbf{p}_{0\ell}^+ \cdots \mathbf{p}_{k\ell}^+ \end{bmatrix} \in \mathbb{R}^{N_\ell \times (k+1)d}$, be a matrix whose column blocks are the first $k+1$ values of the matrix $\mathbf{p}_\ell^+$. We define the matrices $\mathbf{P}_{k\ell}^-$, $\mathbf{Q}_{k\ell}^+$ and $\mathbf{Q}_{k\ell}^-$ in a similar manner with values of $\mathbf{p}_\ell^-$, $\mathbf{q}_\ell^+$ and $\mathbf{q}_\ell^-$ respectively.

Note that except the initial matrices $\{\mathbf{w}_\ell, \mathbf{q}_{0\ell}^-\}_{\ell=1}^L$, all later iterates in Algorithm 3 are random due to the randomness of $\mathbf{V}_\ell$. Let $\mathfrak{S}_{k\ell}^\pm$ denote the collection of random variables associated with the hypotheses, $\mathcal{H}_{k\ell}^\pm$. That is, for $\ell = 1, \ldots, L-1$,

$$\mathfrak{S}_{k\ell}^+ := \left\{ \mathbf{w}_\ell, \mathbf{p}_{\ell-1}^0, \mathbf{P}_{k,\ell-1}^+, \mathbf{q}_\ell^0, \mathbf{Q}_{k\ell}^-, \mathbf{Q}_{k\ell}^+ \right\}, \qquad \mathfrak{S}_{k\ell}^- := \left\{ \mathbf{w}_\ell, \mathbf{p}_{\ell-1}^0, \mathbf{P}_{k-1,\ell-1}^+, \mathbf{q}_\ell^0, \mathbf{Q}_{k\ell}^-, \mathbf{P}_{k,\ell-1}^- \right\}.$$

For $\ell = 0$ and $\ell = L$ we set, $\mathfrak{S}_{k0}^+ := \left\{ \mathbf{w}_0, \mathbf{Q}_{k0}^-, \mathbf{Q}_{k0}^+ \right\}$, $\mathfrak{S}_{kL}^- := \left\{ \mathbf{w}_L, \mathbf{p}_{L-1}^0, \mathbf{P}_{k-1,L-1}^+, \mathbf{P}_{k,L-1}^- \right\}$.

Let $\overline{\mathfrak{S}}_{k\ell}^+$ be the sigma algebra generated by the union of all the sets $\mathfrak{S}_{k'\ell'}^\pm$ as they have appeared in the sequence (37) up to and including the final set $\mathfrak{S}_{k\ell}^+$. Thus, the sigma algebra $\overline{\mathfrak{S}}_{k\ell}^+$ contains all *information* produced by Algorithm 3 immediately *before* line 20 in layer $\ell$ of iteration $k$. Note also that the random variables in Algorithm 4 immediately before defining $P_{k,\ell}^+$ in line 20 are all $\overline{\mathfrak{S}}_{k\ell}^+$ measurable.

Observe that the matrix $\mathbf{V}_\ell$ in Algorithm 3 appears only during matrix-vector multiplications in lines 20 and 32. If we define the matrices, $\mathbf{A}_{k\ell} := \begin{bmatrix} \mathbf{p}_\ell^0, \mathbf{P}_{k-1,\ell}^+ \ \mathbf{P}_{k\ell}^- \end{bmatrix}$, $\mathbf{B}_{k\ell} := \begin{bmatrix} \mathbf{q}_\ell^0, \mathbf{Q}_{k-1,\ell}^+ \ \mathbf{Q}_{k\ell}^- \end{bmatrix}$,

all the matrices in the set $\overline{\mathfrak{S}}_{k\ell}^{+}$ will be unchanged for all matrices $\mathbf{V}_\ell$ satisfying the linear constraints

$$\mathbf{A}_{k\ell} = \mathbf{V}_\ell \mathbf{B}_{k\ell}. \tag{38}$$

Hence, the conditional distribution of $\mathbf{V}_\ell$ given $\overline{\mathfrak{S}}_{k\ell}^{+}$ is precisely the uniform distribution on the set of orthogonal matrices satisfying (38). The matrices $\mathbf{A}_{k\ell}$ and $\mathbf{B}_{k\ell}$ are of dimensions $N_\ell \times (2k+2)d$. From [36, Lemmas 3,4], this conditional distribution is given by

$$\mathbf{V}_\ell|_{\overline{\mathfrak{S}}_{k\ell}^{+}} \overset{d}{=} \mathbf{A}_{k\ell}(\mathbf{A}_{k\ell}^{\mathsf{T}}\mathbf{A}_{k\ell})^{-1}\mathbf{B}_{k\ell}^{\mathsf{T}} + \mathbf{U}_{\mathbf{A}_{k\ell}^{\perp}}\widetilde{\mathbf{V}}_\ell \mathbf{U}_{\mathbf{B}_{k\ell}^{\perp}}^{\mathsf{T}}, \tag{39}$$

where $\mathbf{U}_{\mathbf{A}_{k\ell}^{\perp}}$ and $\mathbf{U}_{\mathbf{B}_{k\ell}^{\perp}}$ are $N_\ell \times (N_\ell - (2k+2)d)$ matrices whose columns are an orthonormal basis for $\mathrm{Range}(\mathbf{A}_{k\ell})^{\perp}$ and $\mathrm{Range}(\mathbf{B}_{k\ell})^{\perp}$. The matrix $\widetilde{\mathbf{V}}_\ell$ is Haar distributed on the set of $(N_\ell - (2k+2)d) \times (N_\ell - (2k+2)d)$ orthogonal matrices and is independent of $\overline{\mathfrak{S}}_{k\ell}^{+}$.

Next, similar to the proof of [36, Thm. 4], we can use (39) to write the conditional distribution of $\mathbf{p}_{k\ell}^{+}$ (from line 20 of Algorithm 3) given $\overline{\mathfrak{S}}_{k\ell}^{+}$ as a sum of two terms

$$\mathbf{p}_{k\ell}^{+}|_{\overline{\mathfrak{S}}_{k\ell}^{+}} = \mathbf{V}_\ell|_{\overline{\mathfrak{S}}_{k\ell}^{+}}\, \mathbf{q}_{k\ell}^{+} \overset{d}{=} \mathbf{p}_{k\ell}^{+\mathrm{det}} + \mathbf{p}_{k\ell}^{+\mathrm{ran}}, \tag{40a}$$

$$\mathbf{p}_{k\ell}^{+\mathrm{det}} := \mathbf{A}_{k\ell}(\mathbf{B}_{k\ell}^{\mathsf{T}}\mathbf{B}_{k\ell})^{-1}\mathbf{B}_{k\ell}^{\mathsf{T}}\mathbf{q}_{k\ell}^{+} \tag{40b}$$

$$\mathbf{p}_{k\ell}^{+\mathrm{ran}} := \mathbf{U}_{\mathbf{B}_{k}^{\perp}}\widetilde{\mathbf{V}}_\ell^{\mathsf{T}}\mathbf{U}_{\mathbf{A}_{k}^{\perp}}^{\mathsf{T}}\mathbf{q}_{k\ell}^{+}. \tag{40c}$$

where we call $\mathbf{p}_{k\ell}^{+\mathrm{det}}$ the *deterministic* term and $\mathbf{p}_{k\ell}^{+\mathrm{ran}}$ the *random* term. The next two lemmas characterize the limiting distributions of the deterministic and random terms.

**Lemma 4.** *Under the induction hypothesis, the rows of the "deterministic" term $\mathbf{p}_{k\ell}^{+\mathrm{det}}$ along with the rows of the matrices in $\overline{\mathfrak{S}}_{k\ell}^{+}$ converge empirically. In addition, there exists constant $d \times d$ matrices $\beta_{0\ell}^{+}, \ldots, \beta_{k-1,\ell}^{+}$ such that*

$$\mathbf{p}_{k\ell}^{+\mathrm{det}} \overset{2}{\Rightarrow} P_{k\ell}^{+\mathrm{det}} := P_\ell^0 \beta_\ell^0 + \sum_{i=0}^{k-1} P_{i\ell}^{+}\beta_{i\ell}^{+}, \tag{41}$$

*where $P_{k\ell}^{+\mathrm{det}} \in \mathbb{R}^{1\times d}$ is the limiting random vector for the rows of $\mathbf{p}_{k\ell}^{\mathrm{det}}$.*

*Proof.* The proof is similar that of [36, Lem. 6], but we go over the details as there are some important differences in the multi-layer matrix case. Define $\widetilde{\mathbf{P}}_{k-1,\ell}^{+} = \left[\mathbf{p}_\ell^0,\ \mathbf{P}_{k-1,\ell}^{+}\right]$, $\widetilde{\mathbf{Q}}_{k-1,\ell}^{+} = \left[\mathbf{q}_\ell^0,\ \mathbf{Q}_{k-1,\ell}^{+}\right]$, which are the matrices in $\mathbb{R}^{N_\ell \times (k+1)d}$. We can then write $\mathbf{A}_{k\ell}$ and $\mathbf{B}_{k\ell}$ from (38) as

$$\mathbf{A}_{k\ell} := \left[\widetilde{\mathbf{P}}_{k-1,\ell}^{+}\ \mathbf{P}_{k\ell}^{-}\right], \quad \mathbf{B}_{k\ell} := \left[\widetilde{\mathbf{Q}}_{k-1,\ell}^{+}\ \mathbf{Q}_{k\ell}^{-}\right], \tag{42}$$

We first evaluate the asymptotic values of various terms in (40b). By definition of $\mathbf{B}_{k\ell}$ in (42),

$$\mathbf{B}_{k\ell}^{\mathsf{T}}\mathbf{B}_{k\ell} = \begin{bmatrix} (\widetilde{\mathbf{Q}}_{k-1,\ell}^{+})^{\mathsf{T}}\widetilde{\mathbf{Q}}_{k-1,\ell}^{+} & (\widetilde{\mathbf{Q}}_{k-1,\ell}^{+})^{\mathsf{T}}\mathbf{Q}_{k\ell}^{-} \\ (\mathbf{Q}_{k\ell}^{-})^{\mathsf{T}}\widetilde{\mathbf{Q}}_{k-1,\ell}^{+} & (\mathbf{Q}_{k\ell}^{-})^{\mathsf{T}}\mathbf{Q}_{k\ell}^{-} \end{bmatrix}$$

We can then evaluate the asymptotic values of these terms as follows: For $0 \leq i, j \leq k-1$ the asymptotic value of the $(i+2, j+2)^{\mathrm{nd}}$ $d \times d$ block of the matrix $(\widetilde{\mathbf{Q}}_{k-1,\ell}^{+})^{\mathsf{T}}\widetilde{\mathbf{Q}}_{k-1,\ell}^{+}$ is

$$\lim_{N\to\infty} \frac{1}{N_\ell}\left[(\widetilde{\mathbf{Q}}_{k-1,\ell}^{+})^{\mathsf{T}}\widetilde{\mathbf{Q}}_{k-1,\ell}^{+}\right]_{i+2,j+2} \overset{(a)}{=} \lim_{N\to\infty}\frac{1}{N_\ell}(\mathbf{q}_{i\ell}^{+})^{\mathsf{T}}\mathbf{q}_{j\ell}^{+}$$

$$= \lim_{N\to\infty}\frac{1}{N_\ell}\sum_{n=1}^{N_\ell}[\mathbf{q}_{i\ell}^{+}]_{n:}[\mathbf{q}_{j\ell}^{+}]_{n:}^{\mathsf{T}} \overset{(b)}{=} \mathbb{E}\left[Q_{i\ell}^{+\mathsf{T}}Q_{j\ell}^{+}\right]$$

where (a) follows since the $(i+2)^{\mathrm{th}}$ column block of $\widetilde{\mathbf{Q}}_{k-1,\ell}^{+}$ is $\mathbf{q}_{i\ell}^{+}$, and (b) follows due to the empirical convergence assumption in (31). Also, since the first column block of $\widetilde{\mathbf{Q}}_{k-1,\ell}^{+}$ is $\mathbf{q}_\ell^0$, we obtain that

$$\lim_{N_\ell\to\infty}\frac{1}{N_\ell}(\widetilde{\mathbf{Q}}_{k-1,\ell}^{+})^{\mathsf{T}}\widetilde{\mathbf{Q}}_{k-1,\ell}^{+} = \mathbf{R}_{k-1,\ell}^{+} \qquad \text{and}$$

$$\lim_{N_\ell\to\infty}\frac{1}{N_\ell}(\mathbf{Q}_{k\ell}^{-})^{\mathsf{T}}\mathbf{Q}_{k\ell}^{-} = \mathbf{R}_{k\ell}^{-}, \tag{43}$$

where $\mathbf{R}^+_{k-1,\ell} \in \mathbb{R}^{(k+1)d \times (k+1)d}$ is the covariance matrix of $\begin{bmatrix} Q^0_\ell \, Q^+_{0\ell} \, \cdots \, Q^+_{k-1,\ell} \end{bmatrix}$, and $\mathbf{R}^-_{k\ell} \in \mathbb{R}^{(k+1)d \times (k+1)d}$ is the covariance matrix of $\begin{bmatrix} Q^-_{0\ell} \, Q^-_{1\ell} \, \cdots \, Q^-_{k\ell} \end{bmatrix}$. For the matrix $(\widetilde{\mathbf{Q}}^+_{k-1,\ell})^\mathsf{T}\mathbf{Q}^-_{k\ell}$, first observe that the limit of the divergence free condition (29) implies

$$\mathbb{E}\left[\frac{\partial f^+_{i\ell}(P^+_{i,\ell-1}, Q^-_{i\ell}, W_\ell, \overline{\Upsilon}^+_{i\ell})}{\partial Q^-_{i\ell}}\right] = \lim_{N_\ell \to \infty} \left\langle \frac{\partial \mathbf{f}^+_{i\ell}(\mathbf{p}^+_{i,\ell-1}, \mathbf{q}^-_{i\ell}, \mathbf{w}_\ell, \overline{\Upsilon}^+_{i\ell})}{\partial \mathbf{q}^-_{i\ell}}\right\rangle = \mathbf{0}, \tag{44}$$

for any $i$. Also, by the induction hypothesis $\mathcal{H}^+_{k\ell}$,

$$\mathbb{E}(P^{+\mathsf{T}}_{i,\ell-1} Q^-_{j\ell}) = \mathbf{0}, \quad \mathbb{E}(P^{0\mathsf{T}}_{\ell-1} Q^-_{j\ell}) = \mathbf{0}, \tag{45}$$

for all $0 \le i, j \le k$. Therefore using (33), the cross-terms $\mathbb{E}(Q^{+\mathsf{T}}_{i\ell} Q^-_{j\ell})$ are given by

$$\begin{aligned}
\mathbb{E}(f^+_{i\ell}(P^0_{\ell-1}, P^+_{i,\ell-1}, Q^-_{i\ell}, W_\ell, \overline{\Upsilon}_{i\ell})^\mathsf{T} Q^-_{j\ell}) &\overset{(a)}{=} \mathbb{E}\left[\frac{\partial f^+_{i\ell}(P^0_{\ell-1}, P^+_{i,\ell-1}, Q^-_{i\ell}, W_\ell, \overline{\Upsilon}^+_{i\ell})}{\partial P^0_{\ell-1}}\right] \mathbb{E}(P^{0\mathsf{T}}_{\ell-1} Q^-_{j\ell}) \\
&\quad + \mathbb{E}\left[\frac{\partial f^+_{i\ell}(P^0_{\ell-1}, P^+_{i,\ell-1}, Q^-_{i\ell}, W_\ell, \overline{\Upsilon}^+_{i\ell})}{\partial P^+_{i,\ell-1}}\right] \mathbb{E}(P^{+\mathsf{T}}_{i,\ell-1} Q^-_{j\ell}) \\
&\quad + \mathbb{E}\left[\frac{\partial f^+_{i\ell}(P^0_{\ell-1}, P^+_{i,\ell-1}, Q^-_{i\ell}, W_\ell, \overline{\Upsilon}^+_{i\ell})}{\partial Q^-_{i\ell}}\right] \mathbb{E}(Q^{-\mathsf{T}}_{i\ell} Q^-_{j\ell}) \overset{(b)}{=} \mathbf{0},
\end{aligned} \tag{46}$$

(a) follows from a multivariate version of Stein's Lemma [23, eqn.(2)]; and (b) follows from (44), and (45). Consequently,

$$\lim_{N_\ell \to \infty} \frac{1}{N_\ell} \mathbf{B}^\mathsf{T}_{k\ell} \mathbf{B}_{k\ell} = \begin{bmatrix} \mathbf{R}^+_{k-1,\ell} & \mathbf{0} \\ \mathbf{0} & \mathbf{R}^-_{k\ell} \end{bmatrix}, \quad \text{and} \quad \lim_{N_\ell \to \infty} \frac{1}{N_\ell} \mathbf{B}^\mathsf{T}_{k\ell} \mathbf{q}^+_{k\ell} = \begin{bmatrix} \mathbf{b}^+_{k\ell} \\ \mathbf{0} \end{bmatrix}, \tag{47}$$

where $\mathbf{b}^+_{k\ell} := \begin{bmatrix} \mathbb{E}(Q^{+\mathsf{T}}_{0\ell} Q^+_{k\ell}) \, \mathbb{E}(Q^{+\mathsf{T}}_{1\ell} Q^+_{k\ell}) \, \cdots \, \mathbb{E}(Q^{+\mathsf{T}}_{k-1,\ell} Q^+_{k\ell}) \end{bmatrix}^\mathsf{T}$, is the matrix of correlations. We again have $\mathbf{0}$ in the second term because $\mathbb{E}[Q^{+\mathsf{T}}_{i\ell} Q^-_{j\ell}] = \mathbf{0}$ for all $0 \le i, j \le k$. Hence we have

$$\lim_{N_\ell \to \infty} (\mathbf{B}^\mathsf{T}_{k\ell} \mathbf{B}_{k\ell})^{-1} \mathbf{B}^\mathsf{T}_{k\ell} \mathbf{q}^+_{k\ell} = \begin{bmatrix} \boldsymbol{\beta}^+_{k\ell} \\ \mathbf{0} \end{bmatrix}, \quad \boldsymbol{\beta}^+_{k\ell} := \begin{bmatrix} \mathbf{R}^+_{k-1,\ell} \end{bmatrix}^{-1} \mathbf{b}^+_{k\ell}. \tag{48}$$

Therefore, $\mathbf{p}^{+\text{det}}_{k\ell}$ equals

$$\begin{aligned}
\mathbf{A}_{k\ell}(\mathbf{B}^\mathsf{T}_{k\ell} \mathbf{B}_{k\ell})^{-1} \mathbf{B}^\mathsf{T}_{k\ell} \mathbf{q}^+_{k\ell} &= \begin{bmatrix} \widetilde{\mathbf{P}}^+_{k-1,\ell} \, \mathbf{P}^-_{k,\ell} \end{bmatrix} \begin{bmatrix} \boldsymbol{\beta}^+_{k\ell} \\ \mathbf{0} \end{bmatrix} + O\left(\frac{1}{N_\ell}\right) \\
&= \mathbf{p}^0_\ell \beta^0_\ell + \sum_{i=0}^{k-1} \mathbf{p}^+_{i\ell} \beta^+_{i\ell} + O\left(\frac{1}{N_\ell}\right),
\end{aligned} \tag{49}$$

where $\beta^0_\ell$ and $\beta^+_{i\ell}$ are $d \times d$ block matrices of $\boldsymbol{\beta}^+_{k\ell}$ and the term $O(\frac{1}{N_\ell})$ means a matrix sequence, $\boldsymbol{\varphi}(N) \in \mathbb{R}^{N_\ell}$ such that $\lim_{N \to \infty} \frac{1}{N}\|\boldsymbol{\varphi}(N)\|^2 = 0$. A continuity argument then shows the empirical convergence (41). $\qquad\square$

**Lemma 5.** *Under the induction hypothesis, the components of the "random" term $\mathbf{p}^{+\text{ran}}_{k\ell}$ along with the components of the vectors in $\overline{\mathfrak{S}}^+_{k\ell}$ almost surely converge empirically. The components of $\mathbf{p}^{+\text{ran}}_{k\ell}$ converge as*

$$\mathbf{p}^{+\text{ran}}_{k\ell} \overset{2}{\Rightarrow} U_{k\ell}, \tag{50}$$

*where $U_{k\ell}$ is a zero mean Gaussian random vector in $\mathbb{R}^{1 \times d}$ independent of the limiting random variables corresponding to the variables in $\overline{\mathfrak{S}}^+_{k\ell}$.*

*Proof.* The proof is identical to that of [36, Lemmas 7,8]. $\qquad\square$

We are now ready to prove Lemma 3.

*Proof of Lemma 3.* Using the partition (40a) and Lemmas 4 and 5, we see that the components of the vector sequences in $\overline{\mathfrak{S}}^+_{k\ell}$ along with $\mathbf{p}^+_{k\ell}$ almost surely converge jointly empirically, where the

components of $\mathbf{p}_{k\ell}^+$ have the limit

$$\mathbf{p}_{k\ell}^+ = \mathbf{p}_{k\ell}^{\mathrm{det}} + \mathbf{p}_{k\ell}^{\mathrm{ran}} \overset{2}{\Rightarrow} P_\ell^0 \beta_\ell^0 + \sum_{i=0}^{k-1} P_{i\ell}^+ \beta_{i\ell}^+ + U_{k\ell} =: P_{k\ell}^+. \tag{51}$$

Note that the above Wasserstein-2 convergence can be shown using the same arguments involved in showing that if $X_N|\mathcal{F} \overset{d}{\Longrightarrow} X|\mathcal{F}$, and $Y_N|\mathcal{F} \overset{d}{\Longrightarrow} c$, then $(X_N, Y_N)|\mathcal{F} \overset{d}{\Longrightarrow} (X, c)|\mathcal{F}$ for some constant $c$ and sigma-algebra $\mathcal{F}$.

We first establish the Gaussianity of $P_{k\ell}^+$. Observe that by the induction hypothesis, $\mathcal{H}_{k,\ell+1}^-$ holds whereby $(P_\ell^0, P_{0\ell}^+, \ldots, P_{k-1,\ell}^+, Q_{0,\ell+1}^-, \ldots, Q_{k,\ell+1}^-)$, is jointly Gaussian. Since $U_k$ is Gaussian and independent of $(P_\ell^0, P_{0\ell}^+, \ldots, P_{k-1,\ell}^+, Q_{0,\ell+1}^-, \ldots, Q_{k,\ell+1}^-)$, we can conclude from (51) that $(P_\ell^0, P_{0\ell}^+, \ldots, P_{k-1,\ell}^+, P_{k\ell}^+, Q_{0,\ell+1}^-, \ldots, Q_{k,\ell+1}^-)$ is jointly Gaussian.

We now need to prove the correlations of this jointly Gaussian random vector are as claimed by $\mathcal{H}_{k,\ell+1}^+$. Since $\mathcal{H}_{k,\ell+1}^-$ is true, we know that (32) is true for all $i = 0, \ldots, k-1$ and $j = 0, \ldots, k$ and $\ell = \ell+1$. Hence, we need only to prove the additional identity for $i = k$, namely the equations: $\mathrm{Cov}(P_\ell^0, P_{k\ell}^+)^2 = \mathbf{K}_{k\ell}^+$ and $\mathbb{E}(P_{k\ell}^+ Q_{j,\ell+1}^-) = 0$. First observe that

$$\mathbb{E}(P_{k\ell}^{+\mathsf{T}} P_{k\ell}^+)^2 \overset{(a)}{=} \lim_{N_\ell \to \infty} \tfrac{1}{N_\ell} \mathbf{p}_{k\ell}^{+\mathsf{T}} \mathbf{p}_{k\ell}^+ \overset{(b)}{=} \lim_{N_\ell \to \infty} \tfrac{1}{N_\ell} \mathbf{q}_{k\ell}^{+\mathsf{T}} \mathbf{q}_{k\ell}^+ \overset{(c)}{=} \mathbb{E}\left(Q_{k\ell}^{+\mathsf{T}} Q_{k\ell}^+\right)^2$$

where (a) follows from the fact that the rows of $\mathbf{p}_{k\ell}^+$ converge empirically to $P_{k\ell}^+$; (b) follows from line 20 in Algorithm 3 and the fact that $\mathbf{V}_\ell$ is orthogonal; and (c) follows from the fact that the rows of $\mathbf{q}_{k\ell}^+$ converge empirically to $Q_{k\ell}^+$ from hypothesis $\mathcal{H}_{k,\ell}^+$. Since $\mathbf{p}_\ell^0 = \mathbf{V}_\ell \mathbf{q}^0$, we similarly obtain that $\mathbb{E}(P_\ell^{0\mathsf{T}} P_{k\ell}^+) = \mathbb{E}(Q_\ell^{0\mathsf{T}} Q_{k\ell}^+)$, $\mathbb{E}(P_\ell^{0\mathsf{T}} P_\ell^0) = \mathbb{E}(Q_\ell^{0\mathsf{T}} Q_\ell^0)$, from which we conclude

$$\mathrm{Cov}(P_\ell^0, P_{k\ell}^+) = \mathrm{Cov}(Q_\ell^0, Q_{k\ell}^+) =: \mathbf{K}_{k\ell}^+, \tag{52}$$

where the last step follows from the definition of $\mathbf{K}_{k\ell}^+$ in line 20 of Algorithm 4. Finally, we observe that for $0 \leq j \leq k$

$$\mathbb{E}(P_{k\ell}^{+\mathsf{T}} Q_{j,\ell+1}^-) \overset{(a)}{=} \beta_\ell^{+\mathsf{T}} \mathbb{E}(P_\ell^{0\mathsf{T}} Q_{j,\ell+1}^-) + \sum_{i=0}^{k-1} \beta_{i\ell}^{+\mathsf{T}} \mathbb{E}(P_{i\ell}^{+\mathsf{T}} Q_{j,\ell+1}^-) + \mathbb{E}(U_{k\ell}^\mathsf{T} Q_{j,\ell+1}^-) \overset{(b)}{=} \mathbf{0}, \tag{53}$$

where (a) follows from (51) and, in (b), we used the fact that $\mathbb{E}(P_\ell^{0\mathsf{T}} Q_{j,\ell+1}^-) = \mathbf{0}$ and $\mathbb{E}(P_{i\ell}^{+\mathsf{T}} Q_{j,\ell+1}^-) = \mathbf{0}$ since (32) is true for $i \leq k-1$ corresponding to $\mathcal{H}_{k,\ell+1}^-$ and $\mathbb{E}(U_{k\ell}^\mathsf{T} Q_{j,\ell+1}^-) = \mathbf{0}$ since $U_{k\ell}$ is independent of $\overline{\mathfrak{G}}_{k\ell}^+$, and $Q_{j,\ell+1}^-$ is $\overline{\mathfrak{G}}_{k\ell}^+$ measurable. Thus, with (52) and (53), we have proven all the correlations in (32) corresponding to $\mathcal{H}_{k,\ell+1}^+$.

Next, we prove the convergence of the parameter lists $\Upsilon_{k,\ell+1}^+$ to $\overline{\Upsilon}_{k,\ell+1}^+$. Since $\Upsilon_{k\ell}^+ \to \overline{\Upsilon}_{k\ell}^+$ due to hypothesis $\mathcal{H}_{k\ell}^+$, and $\varphi_{k,\ell+1}^+(\cdot)$ is uniformly Lipschitz continuous, we have that $\lim_{N \to \infty} \mu_{k,\ell+1}^+$ from line 17 in Algorithm 3 converges almost surely as

$$\lim_{N \to \infty} \left\langle \varphi_{k,\ell+1}^+(\mathbf{p}_\ell^0, \mathbf{p}_{k\ell}^+, \mathbf{q}_{k,\ell+1}^-, \mathbf{w}_{\ell+1}, \overline{\Upsilon}_{k\ell}^+) \right\rangle = \mathbb{E}\left[\varphi_{k,\ell+1}^+(P_\ell^0, P_{k\ell}^+, Q_{k,\ell+1}^-, W_{\ell+1}, \overline{\Upsilon}_{k\ell}^+)\right] = \overline{\mu}_{k,\ell+1}^+, \tag{54}$$

where $\overline{\mu}_{k,\ell+1}^+$ is the value in line 17 in Algorithm 4. Since $T_{k,\ell+1}^+(\cdot)$ is continuous, we have that $\lambda_{k,\ell+1}^+$ in line 18 in Algorithm 3 converges as $\lim_{N \to \infty} \lambda_{k,\ell+1}^+ = T_{k,\ell+1}^+(\overline{\mu}_{k,\ell+1}^+, \overline{\Upsilon}_{k\ell}^+) =: \overline{\lambda}_{k,\ell+1}^+$, from line 18 in Algorithm 4. Therefore, we have the limit

$$\lim_{N \to \infty} \Upsilon_{k,\ell+1}^+ = \lim_{N \to \infty} (\Upsilon_{k,\ell}^+, \lambda_{k,\ell+1}^+) = (\overline{\Upsilon}_{k,\ell}^+, \overline{\lambda}_{k,\ell+1}^+) = \overline{\Upsilon}_{k,\ell+1}^+, \tag{55}$$

which proves the convergence of the parameter lists stated in $\mathcal{H}_{k,\ell+1}^+$. Finally, using (55), the empirical convergence of the matrix sequences $\mathbf{p}_\ell^0, \mathbf{p}_{k\ell}^+$ and $\mathbf{q}_{k,\ell+1}^-$ and the uniform Lipschitz continuity of the update function $f_{k,\ell+1}^+(\cdot)$ we obtain that $\mathbf{q}_{k,\ell+1}^+$ equals

$$\mathbf{f}_{k,\ell+1}^+(\mathbf{p}_\ell^0, \mathbf{p}_{k\ell}^-, \mathbf{q}_{k,\ell+1}^-, \mathbf{w}_{\ell+1}, \Upsilon_{k,\ell+1}^+) \overset{2}{\Rightarrow} f_{k,\ell+1}^+(P_\ell^0, P_{k\ell}^-, Q_{k,\ell+1}^-, W_{\ell+1}, \overline{\Upsilon}_{k,\ell+1}^+) =: Q_{k,\ell+1}^+,$$

which proves the claim (33) for $\mathcal{H}_{k,\ell+1}^+$. This completes the proof. $\qquad \square$

An overview of the iterates in Algorithm 3 is depicted in (TOP) and (MIDDLE) of Figure 3. Theorem 2 shows that the rows of the iterates of Algorithm 3 converge empirically with $2^{\text{nd}}$ order moments to random variables defined in Algorithm 4. The random variables defined in Algo. 4 are depicted in Figure 3 (BOTTOM).