[Reviews · NeurIPS 2020]

Review 1

Summary and Contributions: The authors extend the Multi Layer Vector Approximate Message Passing (ML-VAMP) framework, used for inference of signals in multi-layer Generalised Linear Models, to the matrix-valued signal case, and call the algorithm ML-Mat-VAMP. In addition they provide an asymptotic analysis through the demonstration of so-called state evolution equations, that allow to describe the dynamics of the ML-Mat-VAMP through "macroscopic variables" such as mean-square error etc. They also provide numerical experiments to illustrate the validity of the state evolution in order to test error in simple learning model of a shallow neural net. UPDATE POST AUTHOR RESPONSE: My concerns have been mostly answered. About the main concern: the authors mention they will update the numerical part so that their example will not be covered anymore by the existing theory found in ref [1] which is satisfying. About my unanswered concerns 1) the assumptions behind the validity of the main theorem 1 are not clear enough: are the empirical convergences in display below line 216 assumptions or proved? Is the theorem 1 based on that? Reading again more carefully I realise that is properly stated as assumption on initialisation values, and so are natural. This is not a problem anymore for me, it came from my misunderstanding and too fast reading. About the concern 2) Fig 2: isn't strange that, given a number of train samples, the test error seems higher for higher SNR (comparing the two plots) ? I will assume that it may also come from a misunderstanding from me (and that the authors will comment on this point for clarification in the final version if accepted) or coming from a typo that the authors will correct. To summarize I'm satisfied with the authors answers and maintain my mark.

Strengths: The paper is complete in the sense of providing 1) an interesting inference framework and very nice examples of applications, 2) an associated inference algorithm for matrix-valued signals, 3) its asymptotic analysis and associated rigorous guarantees, 4) a welcomed numerical illustration of the theory. I like very much the section 2 on applications. I did not check the proofs because of lack of time, but they are based on well established tools from the AMP theory. The significance to the NeurIPS community is evident. About the novelty: the paper is slightly incremental because it is an extension of a previously know algorithm (ML-VAMP). Yet extending the framework (algorithm + state evolution) to matrix-valued signals is non-trivial and clearly required a separate publication.

Weaknesses: Apart from the slight incremental aspect of the paper, see my previous comments, I do not see major weaknesses. Another point is: the assumptions behind the validity of the main theorem 1 are not clear enough. Related to that: are the empirical convergences in display below line 216 assumptions or proved? Is the theorem 1 based on that? If they are assumptions as written, the authors should argue in some way why they are valid as they are a bit mysterious, and cite them explicitly as assumptions in the statement of the main theorem.

Correctness: I did not check the proofs because of lack of time, but they are based on well established tools from the AMP theory. The numerical experiments clearly validate the theory too.

Clarity: Overall the paper is very well written (even if quite technical) apart from the theorem assumptions, see my previous comment.

Relation to Prior Work: _The references are not unified: some have name + surname, other have name + intial. Please unify _l 48: in addition of [9,10,3] the authors should also mention the older work by Kabashima who derived AMP in 2003 much before the cited works: "A CDMA multiuser detection algorithm on the basis of belief propagation"

Reproducibility: Yes

Additional Feedback: _eq (3) is not clear to me: do you mean argmin_{Z_0 such that Z_{L-1}=g(Z_0)} g(...) ?? Written as it is the minimisation problem is not clear. _l 83: what means PRS-ADMM ??? _l 95: typo: ...on the rows of F_{k:} of F... -> ...on the rows F_{k:} of F... _l 98: the L2 norm shouldn't be a L1 norm in order to enforce sparsity as usually the case in LASSO? _l 122: even if understandable, w.l.o.g. is not a standard acronym I believe _l 142: the Z inside the log should be bold _l 165: typo : \ell-1 in {Z_\ell}_{\ell=1}^{\ell-1} -> L-1 _l 210: it misses the 1/N in order to define the empirical average no? _l 240: iteartion _in section 5: the M-VAMP used is with the MAP denoiser correct? _l 264: covaraince _Fig 2: isn't strange that, given a number of train samples, the test error seems higher for higher SNR (comparing the two plots) ? Eg at train samples = 4000. Is the SNR actually a inverse SNR ? _l 275: tat _l 74: typo : ...version here can provide be applied to other... _section 5: the model studied here, and presented in sec. 2.2, is similar as the one of reference 1 because the authors chose iid Gaussian weights. So what is the difference / advantage of using ML-Mat-VAMP wrt to the AMP of ref [1] multiple times? Moreover in this committee machine there are no correlations among the columns of the weight matrix that is learned / inferred (correct?). So I do not see why one should use ML-Mat-VAMP instead of multiples times the AMP of ref [1], or am I missing something? I think the authors should discuss this point in any case as confusions are possible here. If I am correct then I think the authors should also think about presenting an example with column-wise correlations where ML-Mat-VAMP presents an advantage wrt the AMP of ref [1].


Review 2

Summary and Contributions: This paper focuses on the inference problem of multi-layer neural networks. The authors have generalized the ML-VAMP algorithm which is designed for vector input to ML-MAT-VAMP which is designed for matrix input.

Strengths: The inference problem of multi-layer neural networks is very relevant to NeurIPS community. The extension to matrix input is a significant extension.

Weaknesses: The main weakness of the paper is the lack of a clear asymptotic assumption. For example, does d (column number of input matrix) grows with sample size or width of layers. What is the order of the width of each layer and the variance of the weights? The main reason I point this out is that under certain asymptotics, the network becomes trivial, for example, suppose the input is in a constant order. When the weights are in 1/w scale with w being the width, then by the law of large number, the output concentrates on a fixed value. When the weights are constant order, then with high probability, the output before entering the activation function is unbounded and can become fixed value after the activation function such as sigmoid. Also, I suspect that the order of d is cruel for the SE holds. In light of this, please discuss the asymptotics clearly in the main paper. Another weakness or maybe a suggestion to improve the significance of the paper is to simplify and analyze the SE of some examples such as those listed in Section 2. The main advantage of the AMP framework is it enables us to analyze the performance of the algorithm and understand the optimization better by analyzing the SE. After response: Thanks for the clarification of the asymptotic assumption. I think the results are meaningful under current settings.

Correctness: Due to the lack of asymptotic assumption, it is hard to follow the proof in appendix and I not sure whether the statements are exactly correct.

Clarity: 'Assumption on initialization, true variables' paragraph includes a very long definition of notation and it is not clear when the assumption on the actual initialization or true variables ends. I suggest the authors make a clear definition of the assumption separately. Also the clarity of the asymptotics are not clear in the main paper.

Relation to Prior Work: The related work part looks good to me.

Reproducibility: Yes

Additional Feedback: If the authors can clarify the asymptotics and the proofs are correct, I would give a score between 6 and 7. If the authors can further provide some non-trivial analysis of SE for certain examples, I would give a score at least 8.


Review 3

Summary and Contributions: This paper proposes to estimate input of the neural network in a certain random large-system limit. The author extend the algorithm from vector-inference problem to matrix-inference problem. The theoretical analysis is provided on the convergency of the algorithm.

Strengths: The strengths of this work include proposing a new method to estimate input in the Large System Limit setting, a theoretical analysis on the convergency of the new algorithm and the formulations of several example applications under this framework. The experimental result is on an example in section 2. ,'Learning the Input Layer of a Two-Layer Neural Network'.

Weaknesses: First, the proposed method is rather incremental upon previous works. It seems to be a new application on multi-layer neural network. The empirical result shows that the new algorithm has good performance on problem 9, but this experiment is really on quite a simple toy domain. Something is confusing about section 2. In section 2, this paper shows that several applications can be formulated under this algorithm. But is there any existing algorithm to formulate these applications and what is the experimental results of the previous algorithms. Since the new algorithm is proposed to estimate the input, what is the advantage of this algorithm compared to previous method. The lack of empirical results and theoretical analysis reduce the novelty of this paper.

Correctness: The adopted methodology seems to be correct. The empirical results seems to be convinced.

Clarity: Satisfactory.

Relation to Prior Work: Yes.

Reproducibility: Yes

Additional Feedback:


Review 4

Summary and Contributions: This paper provides an extention of a recent line of work [11, 29] on inference in multi-layer networks to the case of matrix-valued variables. It derives the corresponding AMP-based algorithm and its state evolution for weights from the random rotationally invariant ensemble. The main contribution is thus to provide an algorithm that has theoretical guarantees for this problem at least for some classes of random weights.

Strengths: This is a solid work that derives an AMP-based algorithm and analyzes it for multi-layer inference with weight matrices being from the random rotationally invariant ensemble. This has been done in [11, 29] for vector values variables, this paper extends these results to matrix values variables where the additional dimension d, is fixed, while in the analysis the number of samples and the widths are large with fixed ratios.

Weaknesses: While the general framework treats a novel case not treated in the existing literature (as far as I know), the experimental section treats a case of iid Gaussian matrices and two layers that was already analyzed with the same approach in Ref. [1]. Thus it the paper would be stronger if it treated a case for which the presented theory (and not an existing one) is actually needed. The existence of previous works, notably [1, 11, 29] makes the result somewhat incremental and unsurprising, but still the technical nature of the derivation is non-trivial so overall I think this is a valuable contribution in a line of work of relevance to the machine learning community.

Correctness: As far as i could tell the method is correct.

Clarity: The paper is technical, and thus not that easily accessible to non-expertns, but given that, I found the paper very clear.

Relation to Prior Work: As mentioned above the ref. [1] already treatst the same case that is presented in the numerical section 5 (unless I missed something) and hence a discussion of this would be appropriate. Ideally a more involved examples - with multiple layers or with non-iid matrices would be preferable.

Reproducibility: Yes

Additional Feedback: The expectations in the M-Vamp SE over the d x d matrices look daunting. It would be useful for reproducibility to provide some more details on the implementation and ideally the code used to produce the figure. Some abbreviations are not defined nor referenced, e.g. KKT. Ref. [21] does not seem to deal with the multi-layer case. In Fig. 2 the range of the test-error seems rather misleading as it starts at 1.0 and not at 0. The authors could comment on why they chose a case where the test error remains this large. ------ i have read the author's answer and I will take their word for providing a numerical validation that for a case not covered in existing work (I forgot that the SE of the present algorithm is indeed different from the one of AMP-based algorithms, but still a non-iid example is needed). I also take their word for providing their code so that readers can test the theory without having to code the rather complex expectations. With these additions that I am assuming will be in the final version I raise my ranking to 7.

[Author Response · NeurIPS 2020]

We thank the reviewers for their comments and remarks. We are also grateful for the errata, missed references, as well as the questions posed by the reviewers which help us explain our results with more clarity.

**Relation to ML-VAMP:** A few reviewers wished that the paper could discuss the significance of the ML-Mat-VAMP method over ML-VAMP from [11, 28, 29]. The ML-VAMP algorithm considers only vector-valued quantities in each layer, while the ML-Mat-VAMP considers matrix-valued unknowns. We show ML-Mat-VAMP can analyze a far broader set of applications including Multi-output GLMs, multi-task learning, not possible with ML-VAMP. Extending the proofs to the matrix case is non-trivial as it requires understanding the interaction between columns in each layer.

Also, in addition to analyzing inference problems, we show that the ML-Mat-VAMP can enable studying learning and generalization error of 2-layer NNs. Remarkably, our analysis can provide exact predictions in this case. Previous AMP methods such as ML-VAMP could only study the generalization error of single-output GLMs [ESAP+20]. The application of ML-Mat-VAMP and generalization error in 2-layer NNs is also non-trivial as it requires an interesting recasting of the learning problem into an inference problem.

**Response to Reviewers # 1 and # 4.** The reviewers raise excellent questions requesting the contrast between this work and [1]. The problem considered in our Section 5 is the same committee machine model from [1], with an important difference that our State Evolution (SE) analysis holds for a far broader class of data matrices – ones which are Rotationally Invariant (not just uncorrelated Gaussian features) – which include correlated non-Gaussian feature vectors leading to poorly conditioned data matrices. Due to the lack of space in the rebuttal document, we are unable to demonstrate this via plots. We shall include these experiments on learning committee machines under correlated non-Gaussian features into the main paper (This case is not explained by the model in [1]). The code for generating the figures and a Python implementation for the ML-Mat-VAMP will be made available on a public github repository. Moreover, our results also holds for the several other multi-layer models detailed in Section 2.

**Response to Reviewer # 2.** Thank you for your remarks. The supplementary material has the full details regarding the proof. If accepted, we are allowed to add 1 extra page in the main paper. Per your suggestions, we would add more details about the assumptions and definitions of asymptotic weak limits discussed in Section 4 as well as a summary of the proof. We would also simplify SE for some models of interest, e.g. those mentioned in Section 2. Regarding the assumptions, the asymptotic results hold in the case where the number of rows $\{N_\ell\}_{\ell=1}^L \to \infty$ such that $\lim_{N_0 \to \infty} \frac{N_\ell}{N_0} = \beta_\ell = \mathcal{O}(1)$, but the number of columns satisfies $d = \mathcal{O}(1)$. When applying this model to analyze learning in 2-layer NNs, this is equivalent to the case with input features $p \to \infty$, number of samples $N \to \infty$ such that $\lim_{N,p \to \infty} \frac{p}{N} = \beta = \mathcal{O}(1)$, and the number of hidden units $d = \mathcal{O}(1)$. To our knowledge, this regime of 2-layer NNs has not been analyzed in the recent papers on *double descent* in wide networks [LXS+19].

**Response to Reviewer # 3.** We thank the reviewer for their comments. It is true that a large body of general purpose solvers are available for the inference tasks considered here (e.g. gradient descent methods for MAP inference and MCMC methods for MMSE inference). However, these methods are notoriously difficult to analyze exactly due to the non-convex nature of the problem and dependencies on various factors such as the step size and initialization. The main benefit of ML-Mat-VAMP is *not* that it out-performs these methods. Instead, the main value is that ML-Mat-VAMP offers rigorous and exact predictions on performance in certain high-dimensional regimes. In addition, we show empirically the fixed points of ML-Mat-VAMP agree with standard methods (e.g. Adam). Hence, the paper provides a tool for predicting the performance of commonly-used methods as well.

**Regarding Broader Impact:** It was our understanding that this section was meant for papers introducing implementations of empirical models trained on large public datasets such as GPT-2,3. However on being pointed out by the reviewers, we realize that our work also serves a broader purpose of bringing interpretability to Neural Network based models which significantly impacts the NeurIPS as well as the broader scientific community. We shall address our thoughts in this regard if our manuscript is accepted.

# References

[ESAP+20] Melikasadat Emami, Mojtaba Sahraee-Ardakan, Parthe Pandit, Sundeep Rangan, and Alyson K Fletcher. Generalization error of generalized linear models in high dimensions. In *ICML*, 2020.

[LXS+19] Jaehoon Lee, Lechao Xiao, Samuel Schoenholz, Yasaman Bahri, Roman Novak, Jascha Sohl-Dickstein, and Jeffrey Pennington. Wide neural networks of any depth evolve as linear models under gradient descent. In *Advances in neural information processing systems*, pages 8572–8583, 2019.


[Meta-Review · NeurIPS 2020]

Out of the four reviews, three are above the acceptance threshold with higher confidence, and the remaining one is below the threshold, although with a weaker confidence. Arguably, the major weaknesses of this paper are: It is somehow incremental, extending ML-VAMP to ML-Mat-VAMP. Also, the numerical example dealt with in Section 5, where F_1 is generated as having iid entries, does not really demonstrate the advantage of ML-Mat-VAMP over ML-VAMP. As for the former point, Reviewers #1 and #4 state that it is still non-trivial. As for the latter, the authors in their response promised to provide results of cases with correlated features. Assuming this to be addressed in the final version, I would like to recommend acceptance of this paper. Upon my own reading of this paper, I noticed that there are several errors in the manuscript. The following is a list of those which I found and no reviewer commented. I would appreciate it if the authors take them into account in preparing the final version. - First of all, the assumption that L is an even number should be stated explicitly. - Equation (3): The fact that one takes argmin of H_L+H_0 with respect to Z_0 should be made explicit. The current expression may read as if one should take argmin of H_L with respect to Z_{L-1}. - Lin 142: The symbol \mathbb{Z}_{L-1} is undefined. (The definition may be \mathbb{Z}_{L-1}={\mathbf{Z}_l}_{l=0}^{L-1}, but there is a distinct symbol \mathbf{Z} introduced in line 165 for the same purpose.) - Line 150: Commas needed: (\mathbf{Z}_{L-1}, \cdots, \mathbf{Z}_0) - Line 156: \mathbf{Z} is undefined at this point. The definition is given in line 165. - Equation (11): \prod_{l=1}^{L-1} might read \prod_{l=1}^{L-2}. - Line 185: the belief density density - Line 195: ML-MAT-VAMP -> ML-Mat-VAMP - Line 206: The period at the end of the sentence is missing. - Line 207: The distribution of the remaining variables (are -> is) - Line 219: satisfy -> satisfies; convergence pointwise -> pointwise convergence - Line 229: requires only require - Line 237: approach to -> approach; G^+(\cdot) and G^+(\cdot) -> G^-(\cdot) and G^+(\cdot) - Line 238: exact an analysis -> an exact analysis - Displayed equation after line 241: \mathbf{G}_l^+ -> G_l^+; \Theta_{kl} undefined. - Line 249: The abbreviation LSL (=large system limit?) undefined. - Line 278: The section number 6 is missing. Other points I would like to mention are: - I guess that the ML-Mat-VAMP and its SE equations reduce to the ML-VAMP and its SE equations, respectively, if specializing the former with d=1. If it is the case then it should be stated explicitly. - In the experimental results it seems that the test errors are not monotonic in the number of training samples. Although I guess that it is due to statistical fluctuation originated from the Monte-Carlo evaluation, as mentioned in line 260, I would appreciate it if the authors improve the evaluation to show more stable results in the final manuscript.